*Method*

# Automated sample preparation with SP3 for low-input clinical proteomics

Torsten Müller[1,2], Mathias Kalxdorf[1,3], Rémi Longuespée[4], Daniel N Kazdal[5] , Albrecht Stenzinger[5] & Jeroen Krijgsveld[1,2,*]

## Abstract

High-throughput and streamlined workflows are essential in clinical proteomics for standardized processing of samples from a variety of sources, including fresh-frozen tissue, FFPE tissue, or blood. To reach this goal, we have implemented single-pot solid-phase-enhanced sample preparation (SP3) on a liquid handling robot for automated processing (autoSP3) of tissue lysates in a 96-well format. AutoSP3 performs unbiased protein purification and digestion, and delivers peptides that can be directly analyzed by LCMS, thereby significantly reducing hands-on time, reducing variability in protein quantification, and improving longitudinal reproducibility. We demonstrate the distinguishing ability of autoSP3 to process low-input samples, reproducibly quantifying 500–1,000 proteins from 100 to 1,000 cells. Furthermore, we applied this approach to a cohort of clinical FFPE pulmonary adenocarcinoma (ADC) samples and recapitulated their separation into known histological growth patterns. Finally, we integrated autoSP3 with AFA ultrasonication for the automated end-to-end sample preparation and LCMS analysis of 96 intact tissue samples. Collectively, this constitutes a generic, scalable, and cost-effective workflow with minimal manual intervention, enabling reproducible tissue proteomics in a broad range of clinical and non-clinical applications.

**Keywords** automation; clinical proteomics; FFPE; low-input; SP3
**Subject Categories** Methods & Resources; Proteomics
**Mol Syst Biol. (2020) 16: e9111**

## Introduction

Mass spectrometry (MS)-based proteomic technologies have matured to allow robust, reliable, and comprehensive proteome profiling across thousands of proteins in cells and tissues. This is the result of parallel developments in mass spectrometric instrumentation that continues to gain speed and sensitivity, in liquid chromatographic technology to separate peptides directly interfaced with MS, and in data analysis pipelines for reliable protein identification and quantification. In addition, various workflows have been developed for comparative analyses across many samples, e.g., using isobaric labels allowing sample multiplexing, or using label-free approaches, and short liquid chromatography (LC) gradients. Collectively, this has propelled proteomic studies in multiple areas of basic and mechanistic biology, using deep and quantitative proteomic profiles to understand spatial and temporal aspects of proteome organization and dynamics in a wide variety of conditions (Schubert *et al*, 2017). In addition, the speed, sensitivity, robustness, and general accessibility of present-day proteomic technologies have an increasing appeal for clinical applications, for various reasons: (i) Underlying mechanisms of many diseases are still unclear, where proteome-level information will increase mechanistic insight of (patho)physiological processes; (ii) proteins are the primary targets of almost all current drugs, and insight in their function will help to understand how drugs impact on cellular processes; (iii) for many diseases, there is a persistent lack of powerful protein biomarkers for diagnostic, prognostic, or predictive purposes.

Still, successful implementation of proteomics in a clinical environment has not materialized yet, primarily because of additional requirements that need to be met on top of those in a research environment alluded to above (e.g., proteome coverage, sensitivity). This mostly pertains to (i) the ability to analyze many (possibly hundreds) of samples in an un-interrupted fashion in order to achieve sufficient statistical power in patient cohorts, (ii) simplify the workflow, thereby removing the need for personnel with specific technical skills in proteomics, (iii) achieving an acceptable turnaround time from receiving samples to the generation of a complete proteome profile analysis, and (iv) cost-effectiveness of the workflow. Most of these bottlenecks can be resolved simultaneously by automation, avoiding manual handling, and thereby eliminating the risk of error and variability, while at the same time enabling longitudinal standardization irrespective of the number of samples. Although LCMS has nowadays been sufficiently standardized to achieve excellent performance across hundreds of samples (Bache

1 German Cancer Research Center (DKFZ), Heidelberg, Germany
2 Medical Faculty, Heidelberg University, Heidelberg, Germany
3 EMBL, Heidelberg, Germany
4 Department of Clinical Pharmacology and Pharmacoepidemiology, Heidelberg University, Heidelberg, Germany
5 Institute of Pathology, Heidelberg University, Heidelberg, Germany
  *Corresponding author. Tel: +49 6221 421720; E-mail: j.krijgsveld@dkfz.de

et al, 2018), preceding sample preparation is still highly cumbersome, involving multiple steps to extract, purify, and digest proteins before subsequent LCMS. In an ideal scenario, this procedure should be streamlined into an automated pipeline that accepts processing conditions for any sample type, thereby facilitating universal applicability. Despite the range of existing sample preparation methods (Rappsilber et al, 2003; Huynh et al, 2009; Wiśniewski et al, 2009; Kulak et al, 2014; Wu et al, 2014; Guo et al, 2015; HaileMariam et al, 2018; Ludwig et al, 2018), very few satisfy these demands to universally accommodate the different requirements imposed by various clinical tissue types. For instance, blood cells can be lysed under more gentle conditions than fresh-frozen tissue, while formalin-fixed and paraffin-embedded (FFPE) tissue requires harsh detergent-based methods to efficiently extract proteins (Wiśniewski et al, 2013). Many currently available sample preparation methods have demonstrated their great utility in many application areas of proteomics; however, they also come with some drawbacks. For instance, stage tips (Rappsilber et al, 2003), and its derivative iST (Kulak et al, 2014), do not tolerate detergents, thereby restricting their generic use. Other approaches involve extensive handling procedures such as filtration (Wiśniewski et al, 2009; Kulak et al, 2014; HaileMariam et al, 2018), centrifugation (Wiśniewski et al, 2009; Kulak et al, 2014; HaileMariam et al, 2018), precipitation (Wu et al, 2014), and electrophoresis (Huynh et al, 2009) that are difficult to standardize or scale up, or that lead to undesirable sample losses especially in low-input applications.

To solve several of these issues, we have recently introduced single-pot solid-phase-enhanced sample preparation (SP3) for unbiased protein retrieval and purification (Hughes et al, 2014, 2019). The method utilizes paramagnetic beads in the presence of an organic solvent (> 50% ACN or EtOH) to promote protein binding to the beads, allowing extensive washing to eliminate contaminants, including detergents such as SDS and Triton X-100. After release of proteins off the beads in an aqueous buffer, proteolytic digestion produces clean peptides that can be directly injected for analysis by LCMS. Another distinguishing feature of SP3 is its efficient protein recovery, facilitating low-input applications while maintaining deep proteome coverage. The combined characteristics of scalability, tolerance to detergents, speed, and ease of operation, qualify SP3 as a universal methodology that has enabled a wide variety of applications, including those involving "difficult" sample types as diverse as FFPE tissue (Hughes et al, 2016; Erich et al, 2018) and historical bones (Cleland, 2018). In addition, SP3 especially performs well for low-input applications (Sielaff et al, 2017), e.g., allowing the analysis of single human oocytes (Virant-Klun et al, 2016), and microdissected tissue (Dilillo et al, 2017; Pellegrini et al, 2019).

A property of SP3 that has not been fully exploited yet is the paramagnetic nature of the beads, which allows automation of the procedure on a robotic liquid handling platform. In genomics, automated sample preparation using magnetic beads has been introduced already several years ago (Fisher et al, 2011) and is now commonly used for library generation through kits available from many vendors. Automated sample preparation is far less common in proteomics and is restricted to specific segments in proteomic workflows such as the enrichment of specific sub-proteomes (e.g., AssayMap to purify phosphorylated peptides; Murillo et al, 2018) or for protein digestion and peptide clean-up (Kuras et al, 2019), or to

cases where detergents can be avoided (iST, on an automated system to process plasma and cell lysates; Geyer et al, 2016).

In this study, we implemented SP3 on a liquid handling Bravo system (Agilent Technologies, USA), in order to build a generic, automated, and scalable 96-well format proteomic pipeline that performs all handling steps starting from a tissue lysate and delivering protein digests that can be directly analyzed by LCMS. We verified performance stability over a period of several weeks and demonstrated the ability to reproducibly handle low-input samples, down to low ng samples containing 100 cells or less. To demonstrate the integration of automated SP3 (autoSP3) in a realistic clinical scenario, we analyzed a cohort of pulmonary adenocarcinoma (ADC) samples, successfully associating pathological tumor growth patterns that have strong prognostic implications (Warth et al, 2012) with distinct proteomic signatures. Finally, we combined autoSP3 with ultrasonication for the seamless integration of tissue lysis and protein extraction with downstream processing by SP3, generating peptides from 96 tissue samples simultaneously without manual handling steps. Collectively, this provides an attractive and cost-effective solution for routine, comprehensive clinical studies, easing the introduction of translational proteomic research with minimal hands-on time and low sample consumption.

# Results

## Establishing a generic, automated proteomic sample preparation pipeline

SP3 method is a fast and simple procedure for unbiased clean-up of proteins and peptides from a wide variety of sample types, and its foundation on paramagnetic bead technology should render it adaptable to robotic automation. Here, we implemented SP3 on the Agilent Bravo liquid handling system, which is widely available to many laboratories. Doing so required optimization of a number of steps, including the positioning of required consumables, reagent, and waste volumes, as well as the Bravo accessories, such as magnet, shaker, and heating block to ensure accessibility of all the required components for each consecutive task, such as tips-on, liquid aspiration and dispensing, and tips-off, including the required volumes for reagents, buffers, or waste. An overview of the deck setup is shown in Fig 1A. As a result, the available deck space and the range of motion of the Bravo pipetting head were exploited to fully automate the process starting from cell or tissue lysates to peptides ready for MS analysis for 96 samples simultaneously.

To evaluate autoSP3, we used lysed HeLa cells as input for the core protocol that we developed to execute all steps from protein clean-up to protein digestion (Protocol C; Fig 1A and C, Appendix Protocol C). This was preceded by off-deck reduction and alkylation of proteins with TCEP and CAA for 5 min at 95°C in a PCR thermocycler. Alternatively, reduction and alkylation can be performed as an integral part of the automated process (Fig 1C), as we used for processing of tissues.

AutoSP3 begins with the aliquoting of paramagnetic bead suspension to each sample, spotting 5 µl as a droplet at the wall of each well which is then gently moved into the sample solution by agitation in an orbital shaking accessory. To promote protein binding, we used ACN (Hughes et al, 2014) rather than EtOH (Hughes et al,

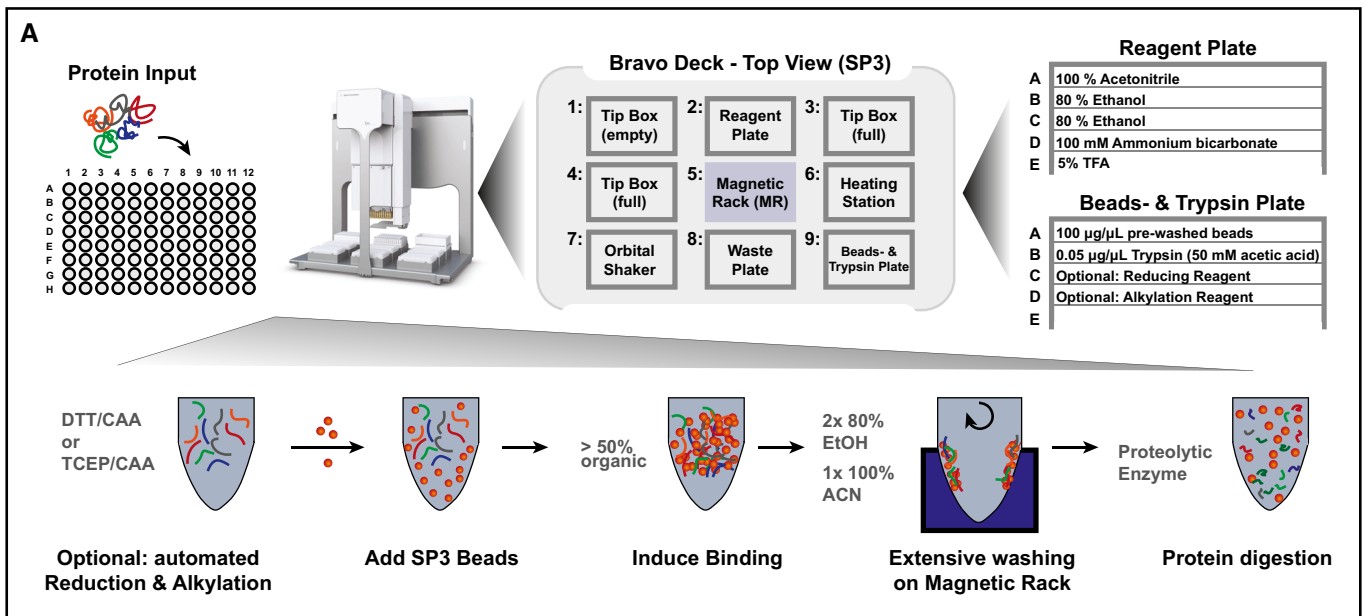

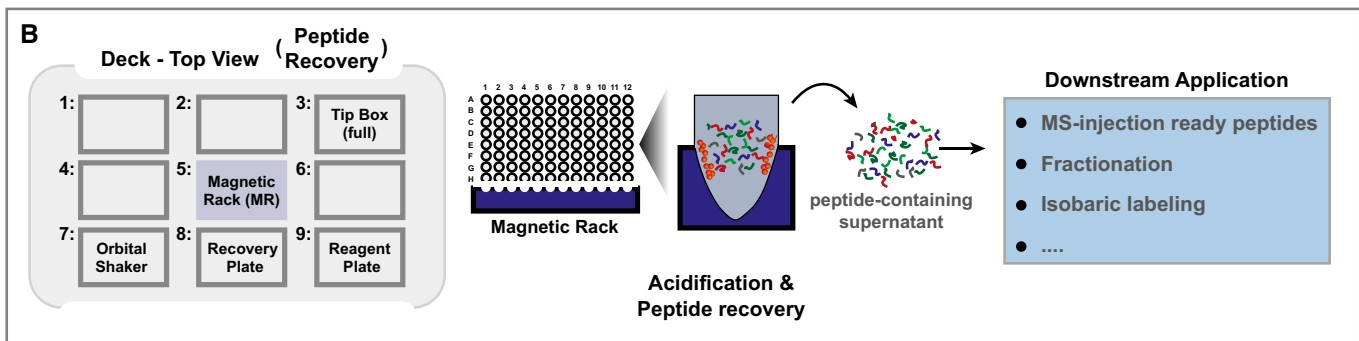

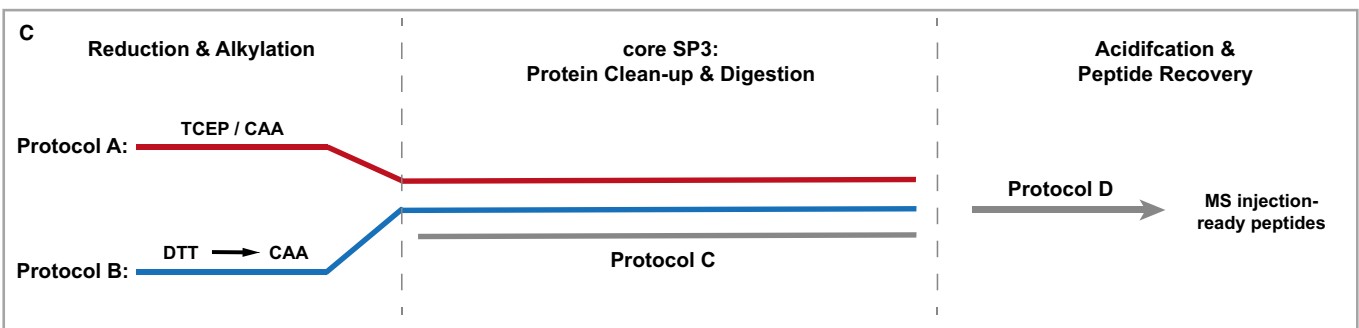

**Figure 1.  A schematic overview of automated single-pot solid-phase-enhanced sample preparation (autoSP3) workflows.**

A   The overview shows the different steps of the autoSP3 protocol from protein input to enzymatic digestion. The setup of the Bravo deck is shown for the core clean-up protocol.

B   The overview shows the steps and Bravo deck setup of the autoSP3 peptide acidification and recovery protocol. The protocol ends with MS injection-ready peptide samples.

C   A schematic overview of all available autoSP3 protocol versions. The autoSP3 procedure is provided with three options for reduction and alkylation and with post-digestion peptide recovery as described in the Appendix Protocols (A to D). Protocol A: one-step reduction and alkylation using a TCEP/CAA mixture for 5 min at 95°C, followed by autoSP3. Protocol B: two-step reduction and alkylation using DTT and CAA consecutively with 30 min of incubation at 60 and 23°C, respectively, followed by autoSP3. Protocol C: the core autoSP3 protocol omitting reduction and alkylation such that the user can flexibly pre-treat manually prepared samples. Protocol D: post-digestion acidification and recovery, delivering MS injection-ready peptides to a new sample plate.

2019) because its properties allow more reproducible pipetting without releasing droplets from the pipette tips. Continuous switching between fast and slow agitation maintains a sufficient distribution of beads by preventing sedimentation and facilitating the efficient formation of protein-bead aggregates. At the same time, this pre-empts pipette mixing and the associated risk of losing sample by beads that tend to stick to the inner wall of the pipette tips under these conditions. The rinsing of beads is performed as described in Hughes *et al* (2019), with two times 80% EtOH and one time 100% ACN. The effective removal of wash solvents within each task is achieved by dividing each liquid aspiration task into two consecutive steps, in which the latter aspirates an additional air plug to avoid hanging droplets. Furthermore, specific liquid classes with optimized aspirating and dispensing velocities were defined for optimal movements. Thus, the complete clearance of all residual solvents is achieved, for example, taking into account the propensity of ACN and especially EtOH to drain from the side wall in each well. Proteins trapped on the paramagnetic beads are resuspended in trypsin (or any other enzyme of choice) and incubated for two to 16 h (Fig 1A). Following enzymatic digestion, peptide samples can be acidified and recovered on-deck in a new plate after supplying new pipette tips (Fig 1B, Appendix Protocol D, or this can be performed manually.

In further optimization steps, the possibility of re-using tips for specific tasks was explored to increase sample throughput and reduce cost. Therefore, we adjusted the liquid dispensing heights in every task such that pipette tips never touch the sample surface or protein-bead aggregates. Thus, it was possible for every liquid-adding task to aspirate sufficient volume only once and successively dispense row-by-row across the entire 96-well plate. Subsequently, during any wash-disposal task, in which the pipette tips inevitably have to dip into the sample solution, aspiration velocities and heights were again optimized to allow liquid transfer without beads sticking to the pipette tips. In addition, the same tip was re-used specifically for the same well in any liquid disposal task, thus excluding the risk of cross-contamination.

Having the optimized workflow in place, we first confirmed that autoSP3 is equally efficient as the established manual procedure with respect to obtained ion intensities as well as the number of identified peptides and proteins from HeLa samples that were processed in parallel (Appendix Fig S1). Next, we verified that cross-contamination in autoSP3 is negligible, by processing 10 μg HeLa protein samples alternating with empty controls across half a 96-well plate. A subset of seven peptide-containing samples and eleven empty controls was randomly selected and subjected to direct LCMS data acquisition, as shown in Appendix Fig S2A and B. Compared to MS intensities in sample-containing injections, most of the empty injections had a residual intensity of < 0.03%, and in all cases well under 1% (Appendix Fig S2B). This could be primarily attributed to autolytic peptides of trypsin (which was added to all samples, including empty ones), and to (non-peptidic) contaminants with a +1 charge state, sharply contrasting with rich chromatograms from protein-containing samples (Appendix Fig S2C).

In summary, we established and optimized the SP3 protocol on a Bravo liquid handling system, taking care of all sample handling steps starting from 96 cell or tissue lysates and producing peptides ready for analysis by LCMS. The estimated cost for processing of 96 samples including reduction, alkylation, and peptide recovery is 92.39 euros (< 1 euro per sample), including magnetic beads (600 μl of 50 μg/μl), trypsin (576 μl of 0.05 μg/μl), reagent and waste plates, three PCR plates, three pipette tip boxes, and all other buffers/solvents.

### Precision of automated SP3 (autoSP3)

A distinguishing feature of any automated procedure is strict standardization leading to precise and reproducible workflows. We evaluated this for automated SP3 by assessing its precision [defined by the EMEA as the variability observed within the same laboratory (EMEA, 2009)], both within the time span of 1 day (intra-day precision) and longitudinally over the period of 1 month (inter-day precision) (Grant & Andrew, 2014). To this end, HeLa cells were lysed, DNA and RNA were digested, and proteins were reduced and alkylated before transfer of protein to a 96-well plate for automated SP3 clean-up and digestion as described above. Intra-day precision was assessed by processing 96 times 10 μg protein of a HeLa lysate in the morning and in the afternoon of three different days (days 1, 13, and 27), i.e., over a time span of roughly 1 month, resulting in a total of six 96-well plates and 576 individual samples (Fig 2A). Inter-day precision was inferred by correlating data obtained across the 3 days. Specifically, LCMS was performed on the respective days immediately after completing automated SP3, by randomly selecting five samples from each of the six plates (i.e., 30 samples). In addition, we analyzed the exact same samples in one complete batch, resulting in additional 30 sample injections, to distinguish potential variance as a result of the SP3 processing from fluctuation in longitudinal MS performance. Collectively, this allowed us to evaluate the variability within a 96-well plate, within a day, across several days, as well as with and without potential variation in MS performance. Intensities of identified peptides were highly consistent across all samples with an average Pearson correlation of 0.9 between each sample, both within and across days (Fig 2B).

To assess intra-day precision at the protein level, we filtered the data obtained from the ten samples generated on a single day for proteins that had been identified and quantified with at least 3 valid values, resulting in 2,672, 2,537, and 2,663 proteins with an LFQ value for day 1, day 13, and day 27, respectively (Fig 2C). For each protein, the coefficient of variation (CV) across ten samples was calculated, demonstrating that more than 91% of the proteins quantified within each day had a CV of < 30%, with a median CV of 12.9, 10.9 and 12.2% for day 1, day 13, and day 27, respectively, reflecting highly consistent protein quantification across replicates of sequentially processed sample plates and within days.

Next, we determined inter-day precision of SP3 performance across all 60 datasets and compared this to data from 16 samples prepared by manual SP3. An average of 14,140 peptide spectrum matches and 3,191 proteins (Appendix Fig S3A) was quantified with CVs of 7.1 and 2.1% across samples, respectively. To maximize the number of proteins included in this assessment, we applied the match-between-runs functionality in MaxQuant, increasing the proportion of proteins that have been quantified without missing values from 33.62 to 58.37%. The median and average CVs of the complete list of quantified proteins ($n = 3,750$) across all 60 samples were 18.1 and 20.5%. For further evaluation, we calculated CVs of proteins with an LFQ intensity in at least 3 out of 60 files ($n = 3,688$) or 45 out of 60 files ($n = 2,964$) (Table 1). The latter

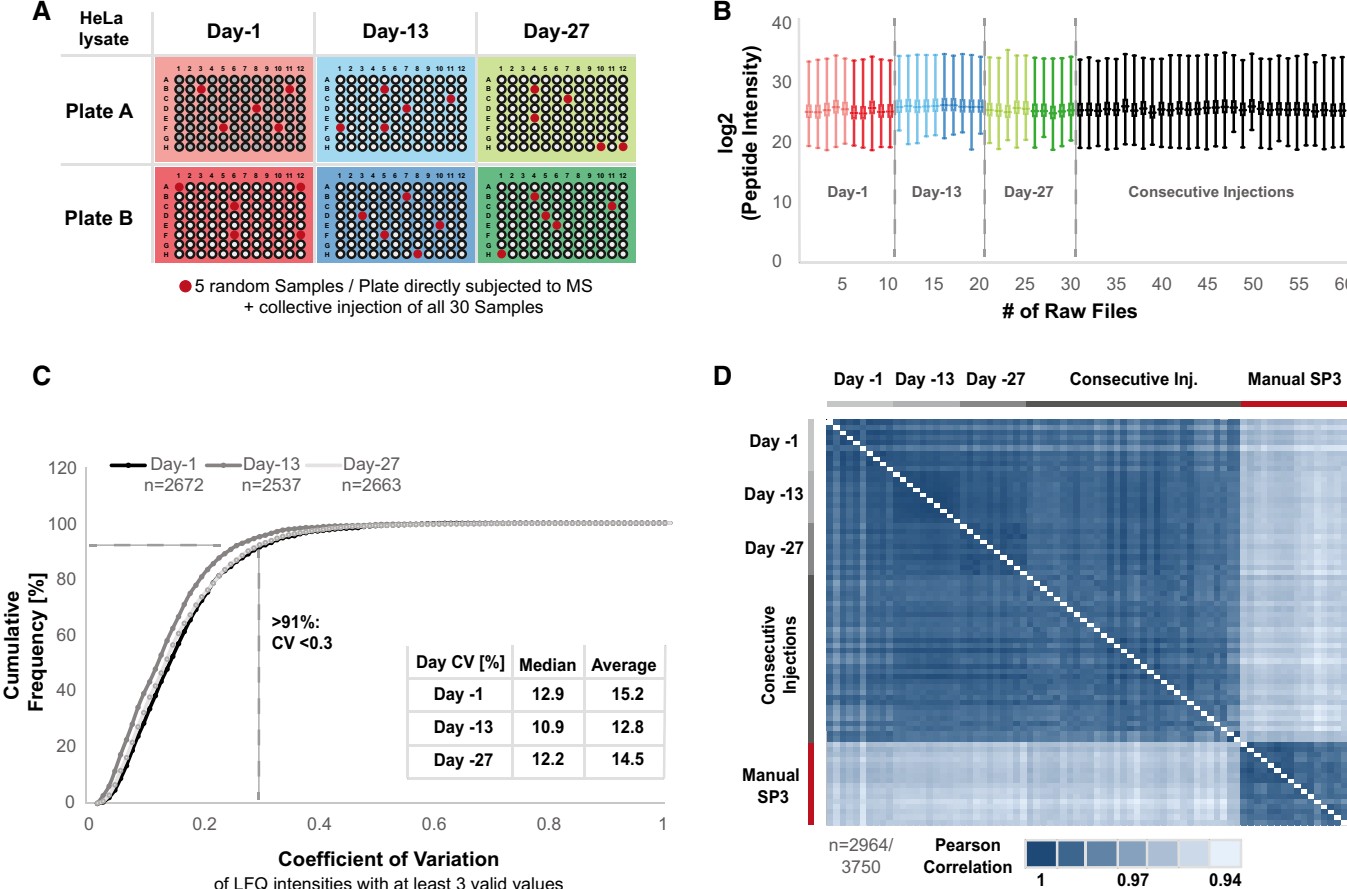

**Figure 2. Evaluation of intra-day precision and inter-day precision.**

A A schematic representation of the experimental design. 96 times 10 µg protein of a HeLa batch lysate were processed in the morning (Plate A) and in the afternoon (Plate B) at three different days (day 1, day 13, and day 27) over a period of a month. From each plate, five randomly selected samples were subjected to direct LCMS analysis (red dots). In addition, all 30 samples (ten per day) were measured in a single combined batch to judge the influence of MS variability

B Box-whisker plots of log₂-transformed peptide intensities across all 60 raw files. The central line of Box-whisker plots represents the median. Boxes extend from the 25th to 75th percentile and whiskers are defined by the smallest and highest value. The color coding highlights the samples plate of origin.

C Cumulative frequency curve [%] of the observed coefficient of variation (CV) of proteins that were identified and quantified with a minimum of three valid values within each day. Here, the ten raw files of each day are evaluated individually. The resulting median and average CV for each day are shown.

D Pearson correlation heatmap of all 60 raw files and additional sixteen manually prepared HeLa SP3 samples. The displayed data are filtered for 75% data completeness (Table 1). Please note the narrow scaling (1–0.94).

Source data are available online for this figure.

corresponds to a minimum data completeness of 75% across all measurements corresponding to 2,964 (79.04%) (Appendix Fig S3B) of the total number of quantified proteins, and it was used for the subsequent comparison of variation within and across different days of sample preparation, as well as with and without the influence of daily LCMS performance (Table 1). CVs across the 30 samples that were analyzed as one batch (median 14.3%; Table 1) were very similar to those analyzed immediately on the day of sample collection (median 13.3%; Table 1), indicating that differences in longitudinal LCMS performance were minimal, and that excellent CVs can be obtained during sample and data acquisition over extended time periods. We observed excellent median and average CVs of proteins across all 60 measurements (14.7 and 17.4%, respectively) showing a marginal but noticeable improvement as compared to the manually processed samples at median CV 16.3% and average CV 18.6%

(Table 1). This is further illustrated in Appendix Fig S4A and B, showing consistently improved CVs in automated versus manual SP3 on a per-protein basis.

Pearson coefficients showed a very high correlation (> 0.97) among both the 60 automatically and 16 manually processed samples, indicating highly robust performance by either procedure (Fig 2D). In addition, no differences are observable between data obtained on days 1, 13, and 27 indicating extremely high inter-day precision. Only slightly lower correlation (> 0.94) was observed between data from manual or automated SP3, reflecting the high robustness of the SP3 protocol itself, but likely reflecting subtle differences between both protocols (e.g., sample volumes).

In addition, we sectioned all proteins identified across 60 experiments in four abundance bins from highest to lowest intensities (A: 1–500; B: 501–1,251; C: 1,252–2,001; D: 2,002–2,964) and

**Table 1. Summary of observed coefficient of variations (CVs).**

| LFQ intra- and inter-day variability | No. of replicates | 75% data completeness (*n* = 2,964) | | Minimum of 3 valid values (*n* = 3,688) | |
|---|---|---|---|---|---|
| | | Median CV (%) | Average CV (%) | Median CV (%) | Average CV (%) |
| Within day 1 | 10 | 11.9 | 14.6 | 13.3 | 16.5 |
| Within day 13 | 10 | 9.8 | 12.2 | 11.3 | 14 |
| Within day 27 | 10 | 10.8 | 13.5 | 12.3 | 15.5 |
| Across days | 30 | 13.3 | 15.7 | 16 | 18.6 |
| w/o MS variability | 30 | 14.3 | 17.3 | 17.2 | 20.1 |
| Overall automated | 60 | 14.7 | 17.4 | 18.1 | 20.6 |
| Manual SP3 | 16 | 16.3 | 18.6 | 17.3 | 20 |

Corresponding to Fig 2, the table summarizes median and average coefficient of variation (CV) values for individual days, across days, with and without the MS imposed variability, and manual SP3. CV values were calculated with either 75% data completeness requirement (~80% of available quantified proteins) or with a minimum of three valid values across 60 samples.

investigated CVs of their LFQ intensities (Appendix Fig S5A). In the two highest abundance bins (A and B), > 97.5% of the proteins have a CV < 30%, with a median well under 10% (Appendix Fig S5A). In the lowest abundance bin (D), only 39.1% of proteins have a CV of < 30%, which, however, comprises the group of ~1,000 proteins that were recovered by the match-between-runs option in MaxQuant. Without this, this number rises to 76.2% (Appendix Fig S5B), while the median CV improves from 34.9 to 25%. Correspondingly, we looked at nine previously described housekeeping proteins (Eisenberg & Levanon, 2013) and two randomly selected proteins with even lower abundance to check CVs at the individual protein level. This demonstrated that CVs both within and across days were well below 5% for the most abundant proteins and below 25% even for the low abundance proteins (Appendix Fig S5C, Table 2).

In conclusion, we have demonstrated robust performance of the SP3 method, irrespective of manual or automated processing, with slightly better median CVs for autoSP3. This comes with additional benefits of high throughput, minimal hands-on time, and highly reproducible longitudinal performance over a period of several weeks.

## Lower limit of processing capabilities of the autoSP3 setup

A persistent challenge in proteomics is the consistent and sensitive analysis of low-input samples. Therefore, we aimed to investigate whether automated SP3 was capable of handling sub-microgram amounts of protein as input material, as we showed before for manual SP3 (Hughes *et al*, 2014; Virant-Klun *et al*, 2016). Therefore, we prepared a 96-well plate with four replicates of twofold serial dilutions of a standard HeLa protein stock, ranging from 10 μg to ~5 ng (Fig 3A), and processed them by autoSP3. The resulting 48 samples (twelve protein concentrations à four replicates) were injected for LCMS in blank-interspaced blocks of replicates from the lowest to the highest amount of protein. Entire samples were injected, except for the 4 highest concentrations which were maximized to 1 μg (back-calculated from the input) to avoid overloading of the analytical column. As expected, the number of quantified proteins and their summed intensities scaled with increased amounts of material, with narrow error distributions across the entire range indicating reproducible processing

independent of input (Fig 3A). 1 μg injections from the four highest concentrated samples consistently identified > 2,000 proteins, indicative of high similarity in sample recovery off the magnetic beads independent of the amount of sample input. In addition, sub-microgram amounts of starting material were still sufficient to quantify several hundreds of proteins (e.g., 403 and 681 proteins from ~39 ng to ~80 ng, respectively). Strikingly, even from ~5 ng of protein input a median of 200 proteins (*n* = 4) was identified and quantified with an iBAQ value. This indicates very efficient protein capture, clean-up, and release by SP3, as well as digestion and transfer on-column with minimal losses.

In another more realistic scenario of limited input material, we started from small numbers of cells instead of aliquoting from a common lysate. Therefore, HeLa cells were counted and directly transferred to a 96-well plate to create a sample series containing 10,000 to 10 cells (in eight replicates divided between two plates; Fig 3B), estimated to correspond to 1 μg to 1 ng of protein material (assuming 0.1 ng/cell). The cells were lysed in the 96-well plate, DNA digested, and directly transferred to the Bravo platform for autoSP3. Again, the number of proteins and iBAQ values scaled with input, where the 1 μg-sample approached 2,000 protein identifications (Fig 3B) as expected for this amount of input (compare to Fig 3A). Excitingly, even starting from 100 cells we quantified on average 459 proteins (*n* = 8). The observed narrow error bars from eight replicates, processed on two individual 96-well plates, demonstrate that the workflow is highly reproducible from beginning to end of the sample handling and including the autoSP3 method, despite the low input.

In summary, the capability to reproducibly quantify 500–1,000 proteins from an input of 100–1,000 cells in a range below 100 ng input opens the door for multiple applications where sample availability is scarce, yet reaching sufficient depth for meaningful experiments. The ability to do so in an automated fashion removes the challenge of manual handling of such small samples.

## Application of autoSP3 to clinical pulmonary adenocarcinoma (ADC) tumors

We next aimed to verify the performance of autoSP3 in a clinical real-world scenario processing a cohort of FFPE tissues, where SP3

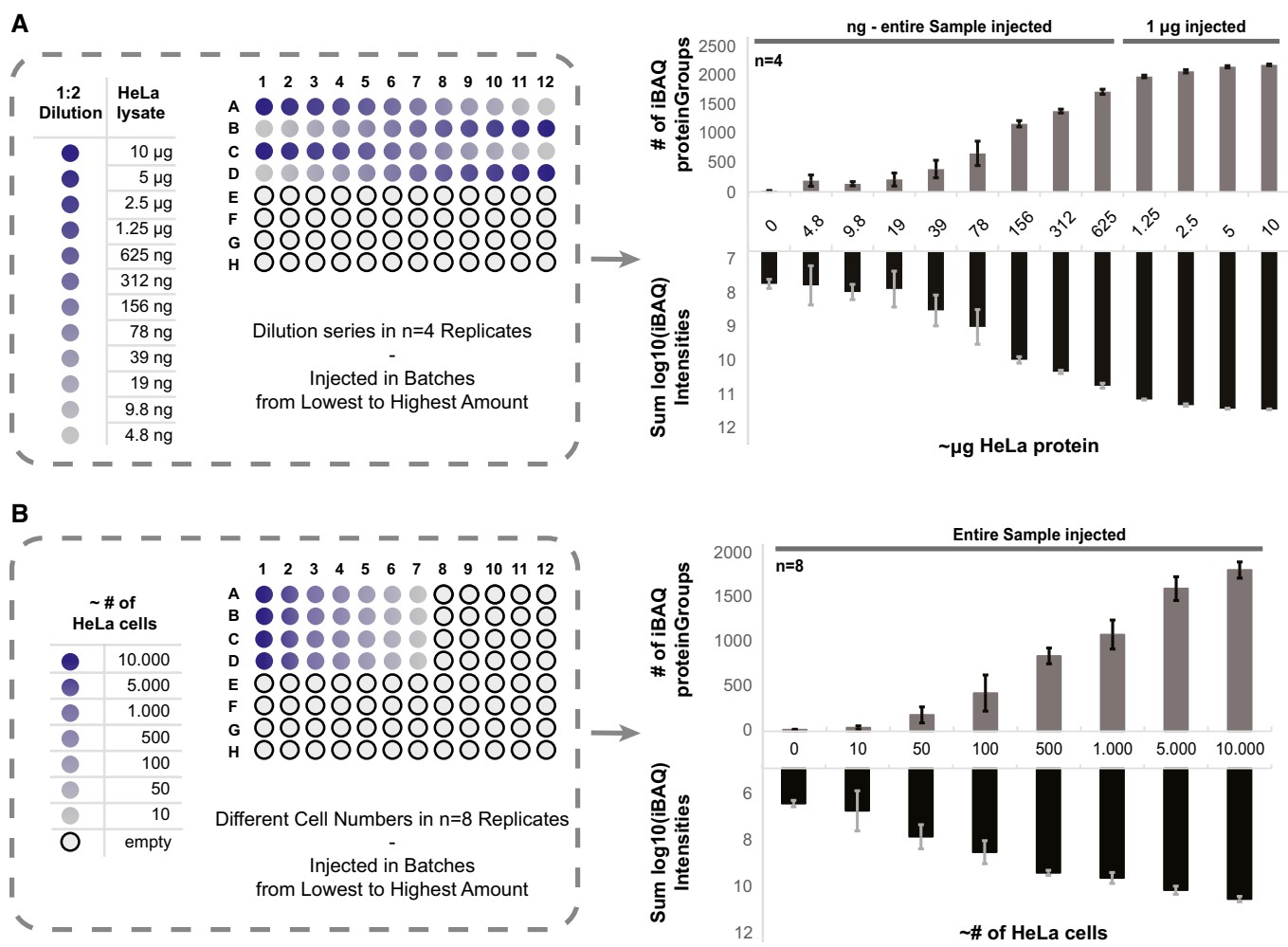

**Figure 3. Evaluation of the lower-limit processing capabilities.**

A   Schematic representation of the experimental design with a 1:2 dilution series of a HeLa batch lysate starting from 10 μg down to 5 ng. The distribution of samples across the 96-well plate is shown. The dilution series was prepared in four replicates, and samples were injected from lowest to highest concentration. For the four highest concentrated samples, 1 μg material was injected, whereas for sub-microgram samples, the entire sample was used. The average number of quantified proteins per sample and the corresponding sum iBAQ intensities are shown with standard deviation error bars from the 4 replicates.

B   Schematic representation of the experimental design of processing low numbers of HeLa cells. Series of decreasing cell numbers were prepared from 10,000 to 10 cells, in eight replicates. The average number of quantified proteins per sample and the corresponding sum iBAQ intensities are shown with standard deviation error bars from the eight replicates.

Source data are available online for this figure.

is uniquely positioned to efficiently remove SDS used for de-crosslinking of this type of samples. Specifically, we aimed to understand the proteomic underpinnings of histological growth patterns observed in pulmonary adenocarcinoma. ADC is the most common histological lung cancer subtype accounting for roughly 60% of non-small-cell lung cancers (NSCLC), known for their heterogeneous clinical, radiologic (Ma *et al*, 2018), molecular (Jamal-Hanjani *et al*, 2017; McGranahan & Swanton, 2017; Kazdal *et al*, 2018), and morphological (Cadioli *et al*, 2014) features. Thus far, five distinct histological growth patterns have been recognized by the 2015 World Health Organization (WHO) Classification of Lung Tumors (Travis *et al*, 2015). These growth patterns, which are

reported in any pathology report, have been proposed for tumor grading according to the predominant pattern of a tumor: lepidic (low grade; group 1), acinar and papillary (intermediate grade; group 2), and solid and micropapillary (high grade; group 3) (Fig 4A). Applying this grading system led to the observation of significant differences regarding prognosis (Warth *et al*, 2012) and prediction of benefit from adjuvant chemotherapy (Tsao *et al*, 2015) where patients with lepidic ADC were associated with the most favorable and patients with micropapillary ADC with the worst prognosis. While marked gene expression differences have been identified for lepidic ADCs (Molina-Romero *et al*, 2017), this is not the case for the other subtypes and thus requires further

**Table 2. Summary of observed coefficient of variations (CVs).**

| LFQ intra- and inter-day variability | Housekeeping genes | | | | | | | | | | |
|---|---|---|---|---|---|---|---|---|---|---|---|
| | VCP | GPI | SNRPD3 | RAB7A | PSMB4 | PSMB2 | VPS29 | REEP5 | CHMP2A | MAP7D1 | ATAD3B |
| Within day 1 | 2.77 | 2.95 | 13.58 | 5.85 | 10.63 | 9.12 | 12.99 | 10.07 | 11.95 | 17.86 | 23.01 |
| Within day 13 | 3.07 | 3.04 | 5.14 | 4.79 | 4.40 | 5.52 | 4.38 | 9.34 | 12.52 | 13.65 | 14.00 |
| Within day 27 | 3.29 | 5.25 | 5.76 | 6.78 | 9.04 | 10.97 | 11.75 | 8.78 | 6.65 | 23.06 | 26.19 |
| Across days | 3.28 | 3.90 | 11.06 | 7.67 | 8.35 | 9.59 | 10.37 | 10.33 | 11.55 | 20.45 | 20.95 |
| w/o MS variability | 3.58 | 4.48 | 8.83 | 5.95 | 6.22 | 8.25 | 11.05 | 10.17 | 11.59 | 23.59 | 19.18 |
| Overall automated | 3.41 | 4.71 | 9.94 | 6.85 | 9.43 | 9.02 | 10.71 | 11.32 | 11.51 | 21.93 | 25.80 |
| Manual SP3 | 7.14 | 3.88 | 5.16 | 10.45 | 11.35 | 10.01 | 10.37 | 14.12 | 14.29 | 29.74 | 28.85 |

Corresponding to Fig 3B, the table summarizes average coefficient of variation (CV) values of individual selected proteins for individual days, across days, with and without the MS imposed variability, and manual SP3.

investigation to identify novel biomarkers, potential therapeutic targets, or to provide a functional explanation for the different growth patterns. In most invasive ADCs, more than one growth pattern can be seen simultaneously, which further highlights the need to better understand functional differences and clinical implications of histological heterogeneity.

Therefore, we collected FFPE tissue samples (5 mm × 5 mm × 5 μm) in a multiregional approach from central sections of eight ADC that had been histologically analyzed using hematoxylin and eosin (H&E) staining to locate and distinguish the different growth patterns. Two to four growth patterns were selected per tumor, and sections were performed in two consecutive iterations to provide replicates with highest possible similarity, resulting in a total of 51 samples (Fig 4A). The tissue was collected in PCR 8 strips, lysed in two batches, and transferred to a 96-well plate in a randomized fashion for autoSP3 clean-up and protein digestion. Peptide samples were batch-randomized and analyzed by LCMS. Injecting ~25% of each sample resulted in the identification of on average 3,576 proteins (Appendix Fig S6A). A t-distributed stochastic neighbor embedding (t-SNE) analysis perfectly grouped replicate samples together (Appendix Fig S6B), despite their random distribution on the 96-well plate during the processing by autoSP3, as well as batch randomization during data acquisition, highlighting the reproducibility of the workflow.

Since samples grouped per patient (Appendix Fig S6B) and not by growth pattern (Appendix Fig S6C), we applied a linear regression model to reduce batch effects. As a result, a t-SNE analysis now separated the three superordinate groups (Fig 4B). Specifically, lepidic and papillary samples were now clearly separated from the remaining samples, while the replicate iterations were still clustered. In addition, acinar, solid, and micropapillary samples clustered more closely but tend to separate at the superordinate level (green and light blue; Fig 4B). The dissimilarity between lepidic and all other samples was expected based on previous reports, while the division of papillary samples in two distinct subclusters separated from the rest of group 2 (acinar) was surprising (further discussed below). The tumor cell content (TCC) is randomly distributed across samples (Appendix Fig S6D), indicating that their separation in the t-SNE analysis is not or at least not solely driven by the TCC.

To gain insight into the potentially growth pattern-specific proteomes, we performed a Limma-moderated t-statistics differential expression analysis comparing each growth pattern versus all combined other samples (Fig 4C, and Appendix Fig S7). Lepidic tissue against all other samples showed the highest number of differentially expressed proteins (167 proteins, Fig 4D and Appendix Fig S7) as expected from the t-SNE analysis (Fig 4B). A gene ontology (GO)-term enrichment analysis of these proteins showed the enrichment of collagens among proteins that are more abundant in lepidic samples (Appendix Fig S8A), reflecting different composition of the extracellular matrix. Collagens have previously been correlated with lung cancer growth, invasion, and metastasis (Hirai *et al*, 1991; Fang *et al*, 2014). Furthermore, we identified mitochondrial ribosomal proteins (MRPs) which have been reported as a predictor for survival and progression, and thus as a potential prognostic biomarker in NSCLC (Sotgia & Lisanti, 2017). In a GSEA of the same data, we found *metabolism of polyamines* and *glucose metabolism* enriched in all others over lepidic (Appendix Fig S8B). Greater capabilities of polyamine synthesis have previously been correlated with accelerated tumor spread and generally higher invasiveness (Soda, 2011), while glucose absorption and metabolism toward anaerobic pathways are a reported key characteristic of the majority of NSCLC, strongly correlated with higher aggressiveness (Giatromanolaki *et al*, 2017). All of the above findings are in line with the known higher aggressiveness and worse prognosis of group 2/3 (intermediate and high grade) compared with group 1 (lepidic, low grade) (Travis *et al*, 2015).

In the comparison of papillary versus all other samples, we found a significant overexpression of PIGT, a subunit of the glycosylphosphatidylinositol transamidase complex. Its deregulation has been associated with NSCLC compared with small-cell lung carcinoma and normal lung tissue with potential implications in diagnosis, prognosis, and therapeutic intervention (Nagpal *et al*, 2008). A GSEA analysis (Appendix Fig S8C) identified *Golgi-associated vesicle budding*, *intra-Golgi* and *Golgi-to-ER trafficking*, and *retrograde transport at the trans-Golgi network* among the top 10 significantly enriched terms pointing to an involvement of the secretory pathway. Altogether, this suggests extensive interaction with the environment in papillary-specific pathology. Interestingly, a relation of secreted proteins and NSCLC has been discussed previously (Huang *et al*, 2006), however, without differentiating between individual growth patterns.

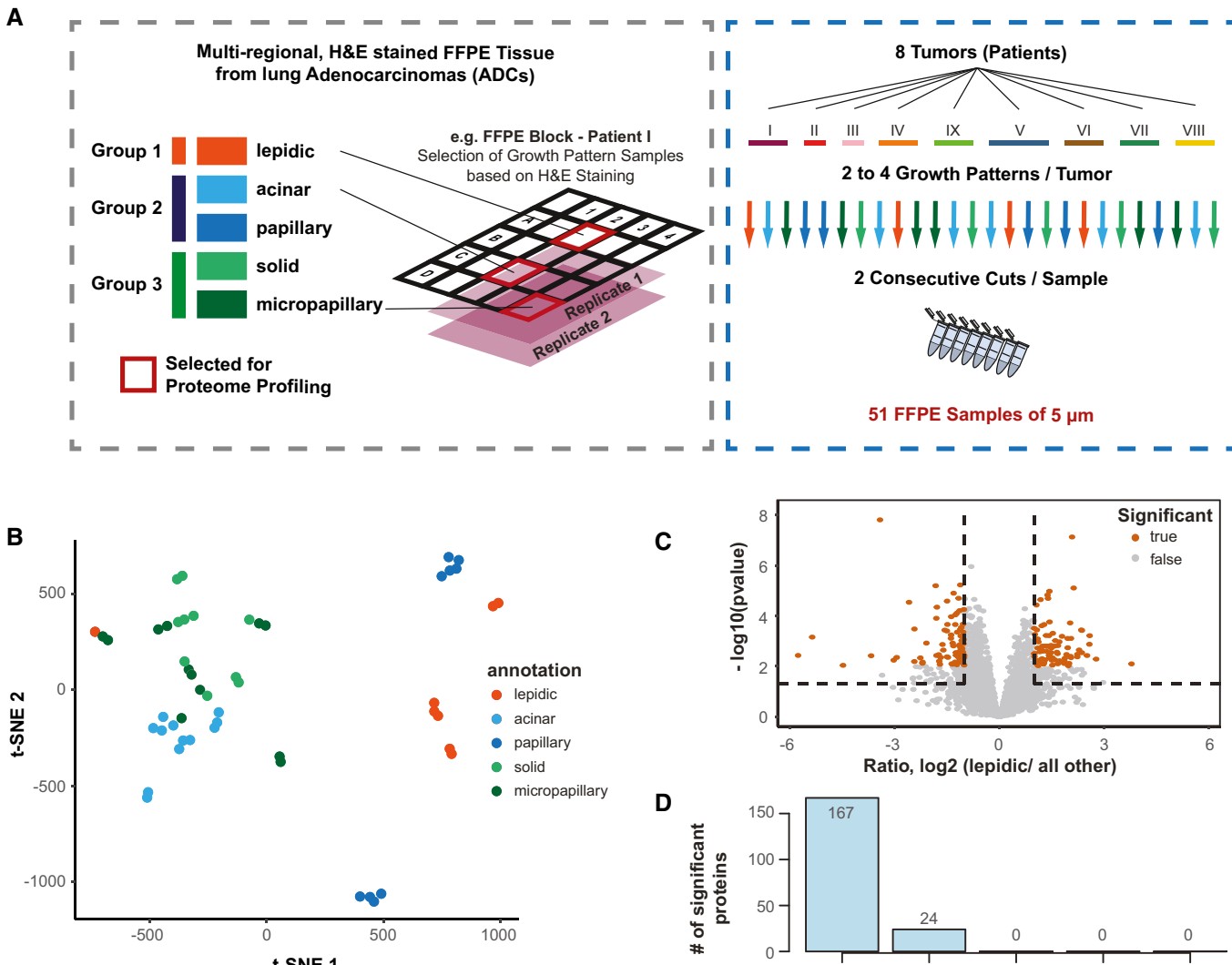

**Figure 4.  Proteome analysis of tumor growth patterns in pulmonary adenocarcinoma (ADC) FFPE tissue.**

A   Schematic illustration of the sample collection. Samples were collected from eight different patient tumors. For each tumor, sections were processed with hematoxylin and eosin (H&E) staining to locate different growth patterns of lepidic (low grade; group 1), acinar and papillary (intermediate grade; group 3), and solid and micropapillary (high grade; group 3). Two to four growth patterns per tumor were selected and sectioned into two consecutive 5 μm iterations, to provide replicates with highest possible similarity, resulting in a total of 51 samples (one iteration was missing).

B   t-Distributed stochastic neighbor embedding (t-SNE) analysis of the proteome data corrected via a linear regression model. The different growth patterns are color-coded as in panel A.

C   Volcano plot showing differential expression analysis using Limma-moderated t-statistics for the comparison of lepidic samples against all other samples. Proteins passing significance thresholds of $-\log_{10} P < 0.05$ (Benjamini–Hochberg adjusted) and an absolute $\log_2$ fold change of $> 1$ are highlighted in orange.

D   Summary of significantly expressed proteins in the comparison of each growth indicated pattern against all others.

Source data are available online for this figure.

We next followed up on the striking observation that papillary samples were separated into two subclusters based on proteome profiles (Fig 4B and Appendix Fig S9A). When subjecting the 73 differentially expressed proteins to a GSEA (Appendix Fig S9B and C), this highlighted collagen-related and extracellular matrix gene sets enriched within Papillary_2 (Appendix Fig S9D), possibly indicating differences in the microenvironment of each of the two papillary subclusters. Processes related to mRNA nonsense-mediated decay

and translation were enriched in the Papillary_1 cluster, indicating differences in the elimination of dysfunctional mRNAs between the subgroups. Further analyses are needed to understand these phenomena in more detail.

In summary, these data demonstrate the applicability of the autoSP3 pipeline to generate quantitative proteome profiles by processing a cohort of clinical ADC FFPE samples. The generated data illustrate high precision by tightly grouping of biological

replicates from randomized samples, and the capability to process FFPE samples for the generation of relevant proteome data revealing differential expression between growth patterns.

## Interfacing ultrasonication with AutoSP3 for automated tissue proteomics

Although the above results demonstrated robust performance of autoSP3, the preceding tissue lysis and reduction/alkylation were performed separately off-deck. To combine all these steps into an integrated workflow that eliminates almost all manual handling steps, we extended the autoSP3 protocol by an automated 96-well format lysis step (Fig 5A). Specifically, we added a LE220R-plus focused-ultrasonicator (Covaris Ltd, UK) to our pipeline, using adaptive focused acoustics (AFA) to dis-integrate tissue and extract proteins by delivery of highly controlled and reproducible energy to small samples volumes (e.g., 12–100 µl) in a 96-well format. In addition, the AFA technology enhances the efficiency of DNA/chromatin shearing to benefit the subsequent autoSP3 protocol.

To demonstrate the seamless integration with the autoSP3 protocol, we lysed 100,000 HeLa cells ($n = 15$) and 1.5–7.5 mg each of three different types of fresh-frozen tissue [pig heart ($n = 16$), mouse liver ($n = 16$), and mouse kidney ($n = 8$)] in the same 96 AFA-tube TPX plate, using a buffer that contains 1% SDS. This takes 5 min per eight samples and linearly scales to 1 h for 96 samples. The amount of protein extracted from each tissue type linearly correlated with the mass of wet tissue input material, liberating ~100 µg protein per mg heart tissue, and ~130 µg per mg liver and kidney tissue, as expected (Hulbert & Else, 1989) (Fig 5B). Upon protein extraction, the 96 AFA-tube TPX plate was directly transferred to the Bravo system for autoSP3, to execute all steps including reduction, alkylation, protein clean-up, digestion, and peptide recovery (using protocols A and D, Fig 1C). Subsequently, we randomly selected five replicates per sample type and continued to acquire proteome data using LCMS. The intensities of identified peptides were highly consistent across all sets of samples with an average Pearson correlation of 0.94, 0.96, 0.95, and 0.87 for heart, liver, kidney, and HeLa cells, respectively (Fig 5C). The number of quantified proteins was highly reproducible across the five replicates per sample type (Fig 5D). To assess the processing precision spanning the entire procedure from tissue lysis to data acquisition by LCMS, we determined the CVs for each sample type (Fig 5E). This demonstrated that more than 84% of quantified proteins have a CV of < 30%, with a median CV of 13.7% (heart), 10.4% (liver), 12.0% (kidney), and 15.5% (HeLa cells), reflecting highly consistent processing across all steps from tissue lysis to generating MS data (Fig 5E).

The complete workflow takes 3.5 h to complete for 96 samples, including 1 h for ultrasonication (tissue lysis, protein extraction, DNA shearing), 2.5 h for autoSP3 (protein reduction, alkylation, and clean-up), and 8 min for peptide recovery. Protein digestion may take 1–16 h, depending on the user's preference. The procedure eliminates manual sample pipetting steps and needs manual intervention only to transfer the sample plate from one station to the next (from the ultrasonicator to the Bravo system, from the Bravo system to a PCR cycler for digestion, back to the Bravo deck for peptide recovery, and to the autosampler for LCMS) (Fig 5A). Collectively, this limits overall hands-on time to < 5 min.

## Discussion

Sample preparation is the only segment in the proteomic workflow that still largely relies on a series of manual handling and pipetting steps, including tissue lysis, protein reduction/alkylation, clean-up to remove contaminating buffer components, and protein digestion. By seamlessly integrating all these steps into an automated process, autoSP3 in combination with AFA-based ultrasonication alleviates many shortcomings that are associated with manual processing. Indeed, we demonstrate excellent reproducibility of the autoSP3 procedure, shown in a series of 60 HeLa samples that were processed spread out over a month's time (median CV of 16.3%), and in a cohort of 51 FFPE samples, where replicate tissue slices originating from subsequent cuts always grouped together despite randomized processing during autoSP3 and ensuing LCMS. Even when including tissue lysis and extraction in the workflow, median CVs remain under 15% (Fig 5E), demonstrating high consistency in all the steps needed to generate peptides from tissues and to analyze these by LCMS. The upshot of this is that samples can be generated over extended periods of time, e.g., during time series or longitudinal tissue collection, without introducing variability due to sample handling. Importantly, all benefits of manual SP3 propagate in autoSP3, including the handling of detergent-containing samples, high sensitivity, and low cost. The ability to handle detergents, including SDS, adds great flexibility to the choice of protein extraction methods and enables processing of sample types that are most efficiently extracted in the presence of SDS, such as FFPE tissue (Wiśniewski *et al*, 2013; Hughes *et al*, 2016). Here, we showed that lysis of both fresh and FFPE tissue is highly efficient in the presence of 1 and 4% SDS, respectively, and that detergent can be effectively removed via autoSP3 without the need for further peptide clean-up before LCMS. The attribute of manual SP3 to perform well with low sample inputs was also demonstrated in autoSP3, showing reproducible identification of 500 proteins from sample amounts as small as 100 HeLa cells (Fig 3). This foreshadows powerful applications for the routine analysis of rare cell types, either in a basic-biological or clinical setting, where, e.g., FACS sorting followed by ultrasonication, autoSP3, and LCMS can be integrated into a streamlined workflow with no other manual intervention than transferring a sample plate from one platform to the next. Importantly, since manual handling of minimal sample amounts is challenging, an automated workflow will reduce technical variability, instead allowing a more insightful focus on biological differences. Finally, the end-to-end autoSP3 workflow is fast, taking 3.5 h to complete for 96 samples (1 h for ultrasonication, 2.5 h for autoSP3) up to the point of digestion. This permits that two to three plates (or up to 300 samples) can be processed per working day by a single operator with minimal hands-on time. Further increase in throughput is easily conceivable, considering that currently 1 h (out of 2.5 h) of the autoSP3 procedure is devoted to heating and cooling of the heating block for reduction and alkylation, which will be considerably faster when using a more efficient device, or by defining conditions where alkylation can be performed at lower temperature. In addition, the use of a plate hotel or increased deck space will increase overall

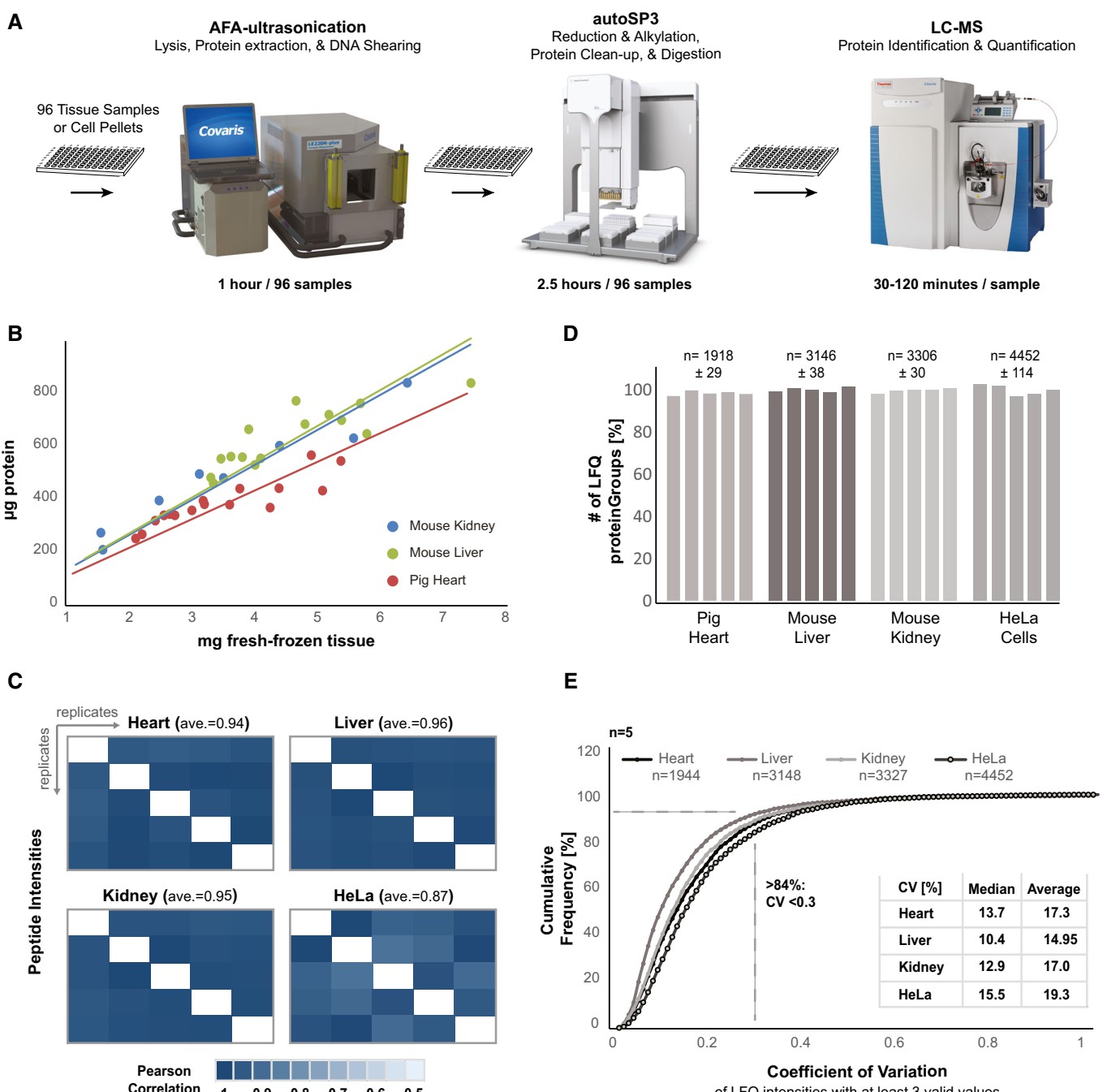

**Figure 5. Interfacing ultrasonication with autoSP3 for automated tissue proteomics.**

A   Schematic illustration of the entire workflow from fresh-frozen tissue or cells to injection-ready peptides and LCMS.

B   Correlation between mg wet tissue weight and the amount of protein extracted by ultrasonication.

C   Pearson correlation heatmap of peptide intensities across five replicates of each sample type (heart, liver, kidney, HeLa cells) with the corresponding average.

D   The relative number of identified and quantified proteins across the five replicates of each sample type. The average number of identified proteins and the standard deviation across five replicates are shown on top.

E   Cumulative frequency curve [%] of the observed coefficient of variation (CV) of proteins that were identified and quantified in at least three out of five replicates in each sample type. The resulting median and average CV for each sample type are shown.

Source data are available online for this figure.

capacity or functionality, e.g., by including a position for protein digestion (in a manner that evaporation is prevented), or TMT labeling (Paulo *et al*, 2019). Finally, the LE220R-plus ultrasonicator has the option to be connected with a robotic arm to facilitate transfer of sample plates to a liquid handling platform, including the Bravo. When implementing one or more of these features in combination, this would generate a hands-free platform that can operate around the clock to process many hundreds of samples per day. This should suffice even for very large-scale proteomic studies, potentially feeding several mass spectrometers.

We have implemented autoSP3 on the Bravo liquid handling system that is widely available in many genomics and biochemistry labs. To facilitate facile adoption, we uploaded all instrument *.vzp files (see Data availability and Appendix Protocols) for the core autoSP3 workflow (Appendix Protocol C) and the extended versions that also include reduction and alkylation (Appendix Protocol A and B), as well as for post-digestion acidification and peptide recovery (Appendix Protocol D). Yet, we expect that SP3 can be readily implemented on other platforms. In this respect, it is interesting to note that our initial protocol for peptide purification by SP3 (Hughes *et al*, 2014) was recently implemented on an (Eppendorf) liquid handling system (Waas *et al*, 2019). This can be an attractive solution for applications beyond protein expression profiling, e.g., to clean-up post-translationally modified peptides after specific enrichment methods. Furthermore, when our paper was under review, a recent study described the use of a KingFisher liquid handling system to perform SP3 in conjunction with subsequent enrichment of phosphopeptides, to facilitate automated sample processing for phosphoproteomic studies (Leutert *et al*, 2019). On the other side of the spectrum, MS analysis of intact proteins purified by SP3 was recently shown (Dagley *et al*, 2019), opening the perspective that the use of autoSP3 might be extended to fit in a workflow for top-down proteomics.

The application of autoSP3 to a cohort of 51 ADC samples demonstrated the ability to process FFPE samples to study molecular differences between tumor growth patterns. In particular, in single-shot LCMS-analyses, we associated decreased expression of proteins involved in cellular invasion in lepidic samples, as expected from pathology. This represented many more lepidic-specific

proteins than previously suggested from gene expression profiles (Molina-Romero *et al*, 2017), implying that (i) not all gene expression differences propagated at the protein level, and (ii) that instead proteome differences arise that do not result from mere gene expression changes. For example, microarray gene expression analyses identified 13 genes with specific differential expression in the lepidic histological growth pattern (Molina-Romero *et al*, 2017). In our dataset, we did not quantify proteins corresponding to the list of 13 genes, but identified 167 differentially expressed proteins that could be of interest for follow-up studies for their potential use as biomarkers or therapeutic targets.

Finally, we demonstrated that the combination of multiplexed ultrasonication with autoSP3 is a very efficient way to produce MS-ready peptides from fresh-frozen tissues in an automated fashion. Removing all manual sample handling therefore uniquely positions this approach as a key building block for routine (clinical) proteome profiling. Although we showed proof of principle for fresh-frozen tissue here, an immediate future development should incorporate the capability to process FFPE samples, which will require further optimization to ideally include both deparaffinization and protein extraction in an integrated sonication protocol. Since histology and WHO classification of tumors almost entirely relies on FFPE, this will open the potential for proteome profiling of samples that have been collected over decades. Utility of the pipeline is further augmented by its additional characteristics with regard to robustness, speed, and sensitivity. For example, the capability of rapid sample processing will contribute to fast turn-around times required for clinical decision making, e.g., to adhere to NSCLC international guidelines for genetic analysis (< 10 days) (Lindeman *et al*, 2013). In addition, low-input capabilities of autoSP3 will allow analysis of small biopsies, possibly including specimens currently not accessible for proteomics, or it may allow reduction in biopsy size to achieve higher tumor cellularity and thus specificity of the assay. Finally, robustness of the method will minimize overall technical variability. Thereby, multiplexed ultrasonication combined with autoSP3 constitutes a highly standardized pipeline that should contribute to the identification of biological or clinical determinants in cohorts of dozens or hundreds of samples.

# Materials and Methods

**Reagents and Tools table**

| Reagent/Resource | Reference or source | Identifier or catalog number |
|---|---|---|
| **Experimental models** | | |
| HeLa cells (*H. sapiens*) | ATCC (Wesel, Germany) | ATCC CCL-2 |
| Pulmonary adenocarcinoma (ADC) FFPE specimens | Thoraxklinik at Heidelberg University (NCT; project: # 1746; # 2818) | N/A |
| **Chemicals, enzymes and other reagents** | | |
| Trypsin-EDTA (0.25%) | Life Technologies (Darmstadt, Germany) | 25200056 |
| 100 × glutamine stock solution | Life Technologies (Darmstadt, Germany) | 25030081 |
| Penicillin-Streptomycin (P&S) mix | Life Technologies (Darmstadt, Germany) | 15140122 |
| Fetal Bovine serum (FBS) | Life Technologies (Darmstadt, Germany) | 10270106 |
| | Life Technologies (Darmstadt, Germany) | 11960085 |

**Reagents and Tools table** (continued)

| Reagent/Resource | Reference or source | Identifier or catalog number |
|---|---|---|
| Dulbecco's modified Eagle Medium (DMEM) with high glucose and no glutamine | | |
| Protease inhibitor cocktail (PIC) | Sigma-Aldrich (Steinheim, Germany) | 5056489001 |
| Tris(2-carboxyethyl)phosphine (TCEP) | Sigma-Aldrich (Steinheim, Germany) | C4706 |
| Chloroacetamide (CAA) | Sigma-Aldrich (Steinheim, Germany) | C0267 |
| Benzonase | Merck (Darmstadt, Germany) | 71206-3 |
| Ethanol (EtOH) | Merck (Darmstadt, Germany) | 34852 |
| Acetonitrile (ACN) | Biosolve Chemicals (Dieuze, France) | 0001204101BS |
| Sodium dodecyl sulfate (SDS) | AppliChem (Darmstadt, Germany) | A0675 |
| Phosphate buffered saline (PBS) | Life Technologies (Darmstadt, Germany) | 70011051 |
| Ammonium bicarbonate (ABC) | Fluka Analytical (Munich, Germany) | FL11213 |
| LCMS-grade water | Biosolve Chemicals (Dieuze, France) | 00232141B1BS |
| Trifluoroacetic acid (TFA) | Biosolve Chemicals (Dieuze, France) | 0020234131BS |
| Formic acid (FA) | Biosolve Chemicals (Dieuze, France) | 0006914143BS |
| Sequencing grade modified trypsin | Promega (Madison, WI, USA) | V5111 |
| Paramagnetic beads for SP3 (Sera-Mag Speed Beads A and B) | Fisher Scientific (Schwerte, Germany) | 24152105050250 & 44152105050250 |
| **Software** | | |
| MaxQuant (version 1.5.1.2) | https://www.maxquant.org/ | |
| Perseus (version 1.6.1.3) | https://maxquant.net/perseus/ | |
| R (version 3.5.1) | https://www.r-project.org/ | |
| Limma moderated t-statistics (R package version 3.36.3) | https://support.bioconductor.org/p/6124/ | |
| VWorks Automation Control software | https://www.agilent.com/en/products/software-informatics/automation-solutions/vworks-automation-control-software | |
| R package fgsea (version 1.6.0 | preprint: Sergushichev (2016) https://doi.org/10.1101/060012 | |
| REACTOME pathway database Gene sets using ReactomePA R package (version 1.24.0) | Yu and He (2016) | |
| t-SNE analyses were performed using R package tsne (version 0.1-3) | van der Maaten and Hinton (2008) | |
| **Other** | | |
| LE220R-plus Focused-ultrasonicator | Covaris | 500578 |
| Branson Sonifier | Branson | NA |
| Pierce BCA Protein assay | Thermo Fisher | 23225 |
| 96-well plates, skirted | Thermo Fisher Scientific | AB-2800 |
| X-Pierce film | Sigma Aldrich | Z721646-50EA |
| 8-row reservoir 32 ml/row | Agilent Technologies | 201260-100 |
| 250 µl tips | Agilent Technologies | 19477-002 |
| Orbital shaking station | Agilent Technologies | Variomag Teleshake |
| MAGNUM FLX enhanced universal magnet | ALPAQUA | https://www.alpaqua.com/Products/Magnet-Plates/Magnum-FLX |
| PCR cycler with lid heating (CHB-T2-D ThermoQ) | Hangzhou BIOER Technologies | CHB-T2-D ThermoQ |
| Bravo liquid handling system | Agilent Technologies | https://www.agilent.com/en/products/automated-liquid-handling/automated-liquid-handling-platforms/bravo-automated-liquid-handling-platform |

| Reagent/Resource | Reference or source | Identifier or catalog number |
|---|---|---|
| Acclaim PepMap C18, 5 µm, 100 Å, 100 µm × 2 cm) | Thermo Fisher Scientific | 164564-CMD |
| Acclaim PepMap RSLC C18, 2 µm, 100 Å, 75 µm × 50 cm | Thermo Fisher Scientific | 11342103 |
| Reprosil-Pur Basic C18 for analytical columns | Dr. Maisch GmbH | NA |
| Easy NanoLC 1200 | Thermo Fisher Scientific | NA |
| Q-Exactive HF Orbitrap mass spectrometer | Thermo Fisher Scientific | NA |

## Methods and Protocols

### Cell culture of HeLa cells

HeLa cells were cultured in regular DMEM medium supplemented with 10% fetal bovine serum, 1% of a 100 × penicillin and streptomycin mix, and 1% of 100 × glutamine stock solution (Gibco). Upon establishment of a stable culture, cells were harvested using trypsin and counted using Bio-Rad TC20 automated cell counter. Cell pellets were stored at −80°C until further use.

For showing the use of the Bravo application starting from small numbers of cells, HeLa cells were harvested, counted, resuspended in lysis buffer (1% SDS, 100 mM ABC pH 8.5), and directly transferred to a 96-well plate. The total volume for different numbers of cells was adjusted using lysis buffer (1% SDS, 100 mM ABC pH 8.5). The entire 96-well plate was sonicated in a waterbath for 10 min, followed by Benzonase (~40 Units) enzymatic cleavage of DNA and RNA for 15 min at 37°C. Subsequently, the buffer was adapted to a final concentration of 1% SDS, 100 mM ABC, 10 mM TCEP, and 40 mM CAA including protease inhibitor cocktail (PIC) before incubation for 5 min at 95°C. The plate was allowed to cool to room temperature before it was transferred to the Bravo deck for the SP3 processing as described in the "automated SP3 protocol" section.

### HeLa protein standard preparation

Cell pellets of ~11.9 million cells were resuspended in 1 ml of lysis buffer (1% SDS, 100 mM ABC pH 8.5, and 50 µl 25× PIC) and probe-sonicated for five times 20 s at a frequency of 10% using a Branson Sonifier. Cell lysates were kept on ice in-between cycles to avoid overheating. DNA or RNA contaminants were cleaved using 250 Units of Benzonase for 15 min at 37°C and 750 rpm. Subsequently, the buffer was adapted to a final concentration of 1% SDS, 100 mM ABC, 10 mM TCEP, and 40 mM CAA including protease inhibitor cocktail (PIC) before incubation for 5 min at 95°C (CHB-T2-D ThermoQ, Hangzhou BIOER Technologies). Reduced and alkylated proteins were quantified using a BCA assay and stored at −20°C until further use in manual and automated SP3 processing.

### Pulmonary adenocarcinoma sample collection

All pulmonary adenocarcinoma (ADC) specimens used for this study were obtained from the Thoraxklinik at Heidelberg University and diagnosed according to the criteria of the 2015 WHO Classification of lung tumors at the Institute of Pathology at Heidelberg University (Travis et al, 2015). Tissue procession to formalin-fixed and paraffin-embedded (FFPE) tissue sections was carried out by the tissue bank of the National Center for Tumor Diseases (NCT;

project: # 1746; # 2818) in accordance with its ethical regulations approved by the local ethics committee.

A multiregional sample set consisting of 2–4 samples of eight tumors was constructed as described previously (Kazdal et al, 2017). In short, a formalin-fixed central section of each tumor was segmented into multiple 5 × 5 mm regions according to a Cartesian grid. Ink marks ensured the retention of the original orientation of each segment during sample processing. Tumor regions considered for analysis were selected in accordance with the tumor size (the larger the tumor the more regions), different histological growth patterns, and sufficient tumor cell content ($\geq 10\%$). The histological growth pattern with predominant portion in each segment was determined by an experienced pathologist. For each tumor, two to four different growth patterns were excised. Samples were analyzed in replicates using one 5-µm section after deparaffinization as input, respectively. For deparaffinization, the sections were incubated for 20 min at 80°C followed by three times 8 min of incubation in Xylol and EtOH, consecutively. Finally, the sections were incubated in ddH$_2$O for 30 min before the tissue was scratched off and collected in a well. Replicates were excised as consecutive cuts of the same region having the highest possible similarity.

### SP3 protocol

As a reference, the SP3 protocol was carried out manually as described before (Hughes et al, 2019). In brief, 10 µg of extracted HeLa protein was added to PCR tubes in a total volume of 10 µl lysis buffer (1% SDS, 100 mM ABC pH 8.5). Magnetic beads were prepared by combining 20 µl of both, Sera-Mag Speed Beads A and B (Fisher Scientific, Germany), washed them one time with 160 µl and two times with 200 µl ddH$_2$O, and re-suspend them in 20 µl ddH$_2$O for a final working concentration of 100 µg/µl. Two micro-litre of pre-washed magnetic beads and 12 µl 100% acetonitrile (ACN) were added to each sample to reach a final concentration of 50% ACN. Protein binding to the beads was allowed for 18 min, followed by 2 min of incubation on a magnetic rack to immobilize beads. The supernatant was removed, and beads were washed two times with 200 µl of 80% ethanol (EtOH) and one time with 180 µl of 100% ACN. Beads were resuspended in 15 µl of 100 mM ABC and sonicated for 5 min in a waterbath. Finally, sequencing-grade trypsin was added in an enzyme:protein ratio of 1:20 (5 µl of 0.1 µg/µl trypsin in ddH$_2$O), and beads were pushed from the tube walls into the solution to ensure efficient digestion. Upon overnight incubation at 37°C and 1,000 rpm in a table-top thermomixer, samples were acidified by adding 5 µl of 5% TFA and quickly vortexed. Beads were immobilized on a magnetic rack, and peptides were recovered by transferring the supernatant to new PCR tubes.

Samples were acidified by adding 75 μl 0.1% FA to reach a peptide concentration of approximately 1 μg/10 μl. MS injection-ready samples were stored at −20°C.

## Automated SP3 protocol (autoSP3)

In the automated version of the SP3 protocol, the Bravo system is programmed to process 96 samples simultaneously, carrying out all handling steps including reduction and alkylation of proteins, aliquoting of magnetic beads, protein clean-up by SP3, protein digestion, and peptide recovery. To facilitate flexible adoption suited to the user's needs, we provide these as various protocols (Fig 1C), all available as instrument files (*.vzp) for direct use on the Bravo system: the core SP3 protocol in combination with reduction and alkylation either as a single-step using TCEP/CAA for 5 min at 95°C (Appendix Protocol A) or as a two-step protocol using, DTT and CAA in consecutive 30 min of incubations (Appendix Protocol B); the core SP3 protocol itself (Appendix Protocol C), leaving the option to perform reduction and alkylation off-deck; a short protocol for peptide acidification, performed upon proteolysis to recover peptides in a new 96-well plate for direct analysis by LCMS (Appendix Protocol D). All protocols A, B, and C are designed for a starting sample volume of 10 μl, which can easily be varied in the protocol files to add respective amounts of organic solvent to reach higher than 50% and to remove the resulting volume after protein binding. Next, either protocol A, B, or C (Fig 1C) aliquot 5 μl of a suspension of washed magnetic beads to protein samples was previously collected in a 96-well plate. Different to the manual protocol (bead working concentration 100 μg/μl), the suspension of washed beads is prepared to have a working concentration of 50 μg/μl to allow more robust pipetting. Next, the respective volume of 100% ACN (20 μl in protocol A; 25 μl in B, 15 μl in C) is added to each sample followed by 18 min of incubation off the magnetic rack with cycles of agitation at 1,500 rpm and 100 rpm for 30 and 90 s, respectively. Upon binding of the proteins to the beads, the sample plate is incubated on the magnetic rack (ALPAQUA, MAGNUM FLX enhanced universal magnet) for further 5 min to allow magnetic trapping of beads inside each well. Here, the beads will form a ring at the wall of each well, slightly above the bottom. The removal of any supernatant is performed using well-specific tips in two consecutive steps to ensure complete liquid removal. Next, beads are washed two times with 200 μl of 80% EtOH and one time with 171.5 μl of 100% ACN. Due to the limited 200 μl pipetting volume of the Bravo and the limited reagent space, the respective washing volumes of 80% EtOH and 100% ACN were added in 4 and 7 consecutive steps of 50 μl and 24.5 μl, respectively, with in-between shaking at 500 rpm or 250 rpm for 30 s. Upon removal of residual washing solvents, the beads are resuspended in 35 μl of 100 mM ABC and 5 μl of 0.05 μg/μl pre-prepared trypsin in 50 mM acetic acid to avoid autolysis. Of note, in the dilution series experiments the amount of trypsin was reduced to avoid abundant peptide features resulting from its autolysis. In a final shaking step at 1,500 rpm for 60 s, the trypsin solution is mixed with the sample and the plate is transferred to the heating deck position for incubation at 37°C. Subsequently, the plate was manually sealed with X-Pierce film and transferred to a PCR cycler with lid heating (CHB-T2-D ThermoQ, Hangzhou BIOER Technologies) to avoid condensation during a 4-h incubation at 37°C. Upon digestion, the film is carefully removed and the plate is placed back on the Bravo for peptide recovery (Appendix Protocol D, Fig 1C). The lid sealing is only used during the protein digestion step at 37°C or during long-term storage of peptides in a new sample plate at the end of autoSP3. Alternatively, peptide acidification and recovery can be performed manually by adding 5 μl of 5% TFA solution, sonication in a waterbath for 5 min to swirl the settled beads, and incubation on a magnetic rack for further 2 min. Finally, the peptide-containing supernatant was recovered into a new 96-well plate without transferring the beads. If necessary, samples were either diluted or directly frozen at −20°C until MS acquisition. Optionally peptide quantification assays (Colorimetric assay kit, Thermo Scientific) were carried out using the Bravo liquid handling system.

## Quantitative proteomic analysis of FFPE tissue

For proteomic analysis, 5-μm FFPE tissue sections were collected in stripes of 8 PCR tubes, centrifuged at 15,000× *g* for 10 min to ensure that FFPE slices are at the bottom of the tube, and stored at 4°C until further processing. Next, each tissue section was carefully reconstituted in 20 μl lysis buffer (4% SDS, 100 mM ABC pH 8.5), sonicated at 4°C for 25 cycles of 30 s on and 30 s off in a Pico Bioruptor, and heated for 1 h at 95°C. Samples were spun down and subjected to a second round of sonication and heating. The Pico Bioruptor was equipped with a house-made tube holder, which allows the simultaneous processing of 28 samples. Subsequently, PCR tubes were centrifuged at 15,000× *g* for 3 min and the buffer was adjusted to a final concentration of 1% SDS, 100 mM ABC, 10 mM TCEP, and 40 mM CAA including protease inhibitor cocktail (PIC). Samples were heated for 5 min at 95°C to denature proteins and to reduce and alkylate cysteine residues. Cooled to room temperature (RT) and again centrifuged at 15,000× *g* for 3 min, 10 μl of each sample was further processed by our automated SP3 sample clean-up procedure, as described above. Here, protein digestion was allowed for 16 h overnight before stopping the reaction by acidification to 0.5% with TFA. The peptide-containing supernatant was recovered to a new 96-well plate without transferring the beads. MS injection-ready samples were stored at −20°C, and about 25% of each sample was later used for data acquisition.

## Multiplexed cell and tissue lysis by ultrasonication

HeLa cells were cultured as previously described. Cells were trypsinized, counted, and transferred in 1× PBS to a 96 AFA-tube TPX plate (Covaris Ltd, UK) and centrifuged at 400× *g* for 4 min. The supernatant was carefully removed, and the remaining cell pellet was resuspended in 12 μl 1% SDS, 100 mM ABC pH 8.5 for sonication.

Fresh-frozen tissue of pig heart, and of mouse liver and kidney were cut on dry-ice into small pieces ranging from 1.5 to 7.5 mg wet weight. Subsequently, each tissue piece was topped with 75 μl of 1% SDS, 100 mM ABC pH 8.5 and transferred to a 96 AFA-tube TPX plate. Focused ultrasonication was performed on a LE220R-plus ultrasonicator (Covaris Ltd, UK) with volume-dependent settings that were optimized for < 25 μl and for 75–100 μl (in brackets): 300 s of duration, 200 peak power (or 300), 25% duty factor, 50 cycles per burst (or 200), and average power of 50 (or 75). The dithering parameters were Y ± 1 mm and speed 20 mm/s for both

methods. For solubilization of heart tissue, two to three rounds of ultrasonication were used.

After sonication, the 96 AFA-tube TPX plate was directly transferred to the Bravo liquid handling system for autoSP3. Of the tissue extracts, 10 µg protein was processed as determined by a BCA assay.

### Proteomic data acquisition

For LCMS analysis of HeLa cells, trypsin-digested samples were diluted with Buffer A (0.1% FA in ddH$_2$O) to enable the injection of 1 µg in 10 µl volume. Peptides were separated using the Easy NanoLC 1200 fitted with a trapping (Acclaim PepMap C18, 5 µm, 100 Å, 100 µm × 2 cm) and an analytical column (Acclaim PepMap RSLC C18, 2 µm, 100 Å, 75 µm × 50 cm). The outlet of the analytical column was coupled directly to a Q-Exactive HF Orbitrap mass spectrometer (Thermo Fisher Scientific). Solvent A was 0.1% (v/v) FA, in ddH$_2$O, and solvent B was 80% ACN, 0.1% (v/v) FA, in ddH$_2$O. The samples were loaded with a constant flow of solvent A at a maximum pressure of 800 bar, onto the trapping column. Peptides were eluted via the analytical column at a constant flow of 0.3 µl/min at 55°C. During the elution, the percentage of solvent B was increased linearly from 4 to 5% in 1 min, then from 5 to 27% in 30 min, and then from 27 to 44% in a further 5 min. Finally, the gradient was finished with 10.1 min at 95% solvent B, followed by 13.5 min at 96% solvent A. Peptides were introduced into the Q-Exactive HF Orbitrap mass spectrometer via a Pico-Tip Emitter 360 µm OD × 20 µm ID; 10 µm tip (New Objective) and a spray voltage of 2 kV.

For LCMS analysis of FFPE lung adenocarcinoma and fresh-frozen tissue, about 25 and 10% of each trypsin-digested sample were used for direct injection, respectively. Peptides were separated using the Easy NanoLC 1200 fitted with a trapping (Acclaim PepMap C18, 5 µm, 100 Å, 100 µm × 2 cm) and a self-packed analytical column (ReproSil-Pur Basic C18, 1.9 µm, 100 Å, 75 µm × 40 cm). The C18 material was packed into fused silica with an uncoated Pico-Tip Emitter with a 10 µm tip (New Objective) using a Nanobaum pressure bomb. The outlet of the analytical column was coupled directly to a Q-Exactive HF Orbitrap mass spectrometer. Solvent A was 0.1% (v/v) FA, in ddH$_2$O, and solvent B was 80% ACN, 0.1% (v/v) FA, in ddH$_2$O. The samples were loaded with a constant flow of solvent A at a maximum pressure of 800 bar, onto the trapping column. Peptides were eluted via the analytical column at a constant flow of 0.3 µl/min at 55°C. During the elution, the percentage of solvent B was increased in a linear fashion from 3 to 8% in 4 min, then from 8 to 10% in 2 min, then from 10 to 32% in a further 68 min, and then to 50% B in 12 min. Finally, the gradient was finished with 7 min at 100% solvent B, followed by 10 min 97% solvent A. Peptides were introduced into the Q-Exactive HF Orbitrap mass spectrometer via a Pico-Tip Emitter 360 µm OD × 20 µm ID; 10 µm tip (New Objective) and a spray voltage of 2.5 kV.

In both settings, the capillary temperature was set at 275°C. Full scan MS spectra with mass range $m/z$ 350–1,500 were acquired in the Orbitrap with a resolution of 60,000 FWHM. The filling time was set to a maximum of 50 ms with an automatic gain control target of 3 × 10$^6$ ions. The top 10 most abundant ions per full scan were selected for an MS$^2$ acquisition. The dynamic exclusion list was with a maximum retention period of 60 s. Isotopes, unassigned

charges, and charges of 1 and > 8 were excluded. For MS$^2$ scans, the resolution was set to 15,000 FWHM with automatic gain control of 5 × 10$^4$ ions and maximum fill time of 50 ms. The isolation window was set to $m/z$ 1.6, with a fixed first mass of $m/z$ 120, and stepped collision energy of 28.

### Proteomic data processing

Raw files were processed using MaxQuant (version 1.5.1.2). The search was performed against the human UniProt database (20170801_Uniprot_homo-sapiens_canonical_reviewed; 20214 entries) using the Andromeda search engine with the following search criteria: Enzyme was set to trypsin/P with up to 2 missed cleavages. Carbamidomethylation (C) and oxidation (M)/acetylation (protein N-term) were selected as a fixed and variable modifications, respectively. First search peptide tolerance and second search peptide tolerance were set to 20 and 4.5 ppm, respectively. Protein quantification was performed using the label-free quantification (LFQ) algorithm of MaxQuant. LFQ intensities were calculated separately for different parameter groups using a minimum ratio count of 1, and minimum and average number of neighbors of 3 and 6, respectively. MS$^2$ spectra were not required for the LFQ comparison. On top, intensity-based absolute quantification (iBAQ) intensities were calculated with a logfit enabled. Identification transfer between runs via the match-between-runs algorithm was allowed with a match time window of 0.3 min. Peptide and protein hits were filtered at a false discovery rate of 1%, with a minimal peptide length of seven amino acids. The reversed sequences of the target database were used as a decoy database. All remaining settings were set as default in MaxQuant. LFQ values were extracted from the protein group table and log$_2$-transformed for further analysis. No additional normalization steps were performed, as the resulting LFQ intensities are normalized by the MaxLFQ procedure. Proteins that were only identified by a modification site, the contaminants, and the reversed sequences were removed from the dataset. All consecutive steps were performed in Microsoft Excel, Perseus (version 1.6.1.3), and the software environment R (version 3.5.1). The differential expression analysis of the ADC samples was performed using Limma-moderated t-statistics (R package version 3.36.3) (Ritchie et al, 2015). Here, the technical replicates and the patient-dependent batch effect were taken into account within the applied model. Proteins with a Benjamini–Hochberg-adjusted $P$-value lower than 0.05 and an absolute log$_2$ fold change higher than 1 were considered as significantly changing. The resulting lists of significantly regulated proteins were subjected to a gene ontology (GO)-term enrichment analyses using the STRING: functional protein association network database (Szklarczyk et al, 2015). The entire list of proteins in the dataset ($n = 5,642$) was used as a background for the analysis. We further required highest confidence (minimal required interaction score of 0.9), excluded text mining, and removed disconnected nodes from the network. The gene set enrichment analyses (GSEA) were performed using R package fgsea (preprint: Sergushichev, 2016) (version 1.6.0) with a $P$-value ranking of proteins, gene sets defined by the REACTOME pathway database (R package ReactomePA version 1.24.0) (Yu & He, 2016), the minimum size of gene sets set to 15, the maximum size of gene sets set to 500, and the number of permutations set to 10,000. The t-SNE analyses were performed using R package tsne (van der Maaten & Hinton, 2008) (version 0.1-3) with a perplexity set to 2 and number of iterations set to 5,000. The coefficient of variation (CV)

was calculated with a minimum of three valid values unless otherwise stated (e.g., 75% data completeness requirement).

### Investigation of intra-day precision and inter-day precision

To test the precision of SP3 sample handling, we followed guidelines of the European Pharmacopoeia and the European Medicines Agency for the number of replicates necessary to validate our method (EMEA, 2009; European Directorate for the Quality of Medicines, 2011). Specifically, we validated automated SP3 by an intra-day and inter-day components by processing a total of six 96-well plates with 10 μg protein of a HeLa batch lysate in each well in the morning and in the afternoon of three different days, over a time span of roughly 1 month, resulting in a total of 575 individual samples. Five randomly picked samples per plate (10 samples per day) were selected for direct LCMS analysis on the day of sample generation and a second technical-repeat injection of all 30 samples in a single batch acquisition. The number of samples per plate to be analyzed was chosen as a fair compromise to determine the precision of our sample processing with a reasonable amount of data acquisition time. The selected samples allowed the evaluation of the inter-day precision and intra-day precision while taking different processing times, plates, and buffers into account (robustness). The second technical injection in one batch allowed to evaluate the influence of longitudinal MS performance. Lastly, for the comparison of manual SP3 sixteen times 10 μg protein of a HeLa batch lysate was processed manually at the bench.

### Lower limit of starting material

To evaluate the lower limit of processing capabilities of the Bravo SP3 setup, we generated starting material dilution series as follows: (i) a dilution series of a standard HeLa-extracted protein stock, ranging from 10 μg to ~5 ng in 1:2 dilution steps (10 μg, 5 μg, 2.5 μg, 1.25 μg, ~625 ng, ~312 ng, ~156 ng, ~78 ng, ~39 ng, ~19 ng, ~10 ng, and ~5 ng). The dilution series was generated and processed in four replicates on the same 96-well plate (12 concentrations and $n = 4$); (ii) a dilution series starting from small numbers of counted cells that were directly dispensed in SDS containing buffer (1% SDS, 100 mM ABC pH 8.5) to a 96-well plate, ranging from 10,000 down to 10 cells. Cells were lysed in the plate by sonication in a waterbath, and DNA was digested using 40 U/μl Benzonase nuclease for 15 min at 37°C. Subsequently, a final concentration of 10 mM TCEP, 40 mM CAA, and PIC was added to each sample for reduction and alkylation for 5 min at 95°C. The 96-well plate was directly transferred to the Bravo platform for autoSP3 resulting in injection-ready peptide samples. The dilution series was generated and processed in two plates à four replicate series (7 concentrations and $n = 8$). Here, the European Pharmacopoeia recommends a minimum of three concentrations à three replicates (European Directorate for the Quality of Medicines, 2011). In addition, two empty control injections were performed upfront of the data acquisition of each dilution series. The dilution series were measured in blank-interspaced blocks from lowest to highest concentrated samples to avoid potential carry over between injections. During the data processing of all dilution series, the "match-between-runs" algorithm in MaxQuant was switched off to avoid the transfer of peptide identifications to background ions.

### Assessment of cross-contamination

To assess potential cross-contamination between samples, we processed 24 wells of 10 μg standard HeLa protein stock interspaced with 24 empty controls. Seven peptide-containing samples and eleven empty controls were randomly selected for direct LCMS analysis. The number of samples to be analyzed was chosen as a fair compromise to determine potential carry over between wells during our sample processing with a reasonable amount of data acquisition time.

## Data availability

The mass spectrometry proteomic data have been deposited to the ProteomeXchange (Deutsch *et al*, 2017) Consortium via the PRIDE (https://www.ebi.ac.uk/pride/) (Perez-Riverol *et al*, 2019) partner repository with the dataset identifier PXD014556 (autoSP3 precision and FFPE application, related to Figs 1–4) and PXD015840 (Multiplexed Cell & Tissue Lysis, related to Fig 5). The instrument files (*.vzp files) for the methods on the Agilent Bravo system corresponding to Protocols A to D have been deposited to the dataset identifier PXD014556.

**Expanded View** for this article is available online.

### Acknowledgements
This research project was supported by the Excellence Cluster CellNetworks to J.K. We thank Gertjan Kramer and Nicolas Autret for helpful discussions, and the Tissue Bank of the National Center for Tumor Diseases for excellent technical assistance.

### Author contributions
TM and JK designed the study; TM performed research; TM and MK analyzed data; RL, DNK and AS provided reagents and samples; TM and JK wrote the paper with input from all authors.

### Conflict of interest
The authors declare that they have no conflict of interest.

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
