## [Review Process File · Molecular Systems Biology]

Automated sample preparation with SP3 for low-input clinical proteomics

Torsten Müller, Mathias Kalxdorf, Rémi Longuespée, Daniel N. Kazdal, Albrecht Stenzinger, Jeroen Krijgsveld.

Review timeline:

Submission date:	13 th July 2019
Editorial Decision:	19 th August 2019
Authors' appeal:	3 rd September 2019
Editorial Decision:	4 th September 2019
Revision received:	31 st October 2019
Editorial Decision:	2 nd December 2019
Revision received:	4 th December 2019
Accepted:	5 th December 2019

Editor: Maria Polychronidou

Transaction Report:

1st Editorial Decision

19th August 2019

Thank you again for submitting your work to Molecular Systems Biology. We have now heard back from the three referees who agreed to evaluate your manuscript. As you will see from the reports below, the reviewers raise substantial concerns on your work, which, I am afraid to say, preclude its publication in Molecular Systems Biology.

The reviewers acknowledge that the presented approach is likely to be relevant for the proteomics community. However, they are concerned that the novelty of the method remains limited and think that the nature of the advance seems mostly technical. A rather important concern is raised by reviewer #3, who points out that several steps of the protocol (including lysis, digestion and peptide harvesting) rely on manual processing. Reviewer #3 mentions that in absence of a fully automated protocol, the broader applicability of the approach remains unclear. We think that this concern is quite substantial and in combination with the somewhat limited enthusiasm expressed by the reviewers regarding the overall methodological advance, at this point we see no choice but to return the manuscript with the message that we cannot offer to publish it.

Nevertheless, we acknowledge that, as the reviewers mention, the proposed workflow is likely to be useful for the community. As such, we would not be opposed to considering a revised manuscript based on this work, provided that the concern regarding the automation of the initial steps of the workflow could be addressed. We recognize that addressing this concern would involve substantial further analyses. In case you decide to revise your study along those lines and you would like to share your revision plan with me, I would be happy to discuss further.

REFeree REPORTS

Reviewer #1:

In this manuscript from Mueller and colleagues (hereafter referred to as the Mueller Report), the authors advance their solid-phase protein extraction and digestion method to an automated liquid-handling robot and validate its performance. This "automated SP3" sample preparation is then challenged with low sample inputs (protein from ~100 cells) and a real-world application in profiling protein abundance differences from human patient-derived FFPE samples of lung adenocarcinoma tumors of various histological subtypes.

This is a timely, informative and useful study that is sure to help researchers in the field of quantitative proteomics work more efficiently, reproducibly and easily on their research projects. It's pretty well written, although I have a few minor suggestions to streamline the text a bit. Although I hate to bring up the dreaded "conceptual innovation" topic, it should be clear that while incredibly useful and artfully described and validated, this is not a novel or new idea implemented for the first time. In my view, however, the utility of these methods far outweighs the often overwrought emphasis on novelty.

There is considerable effort taken in the early part of the manuscript to describe things that were attempted to be automated but failed. This reads a little too conversational; if space considerations are important, much of this could be streamlined. A more succinct depiction of the workflow and the automation in the early Results section might be better served in the main text, with anecdotes reserved for maybe Supporting Information, if at all.

To readers not familiar with the initial SP3 publication, comparison of an automated with manual SP3 method may not seem rigorous enough. Some additional discussion (or a simple comparison experiment) on traditional FASP/8M urea/etc. preparation and digestion approaches might be helpful here (even self-citations to the 2014 paper). For example, one might expect 1ug of HeLa cell digest analyzed on a Q-Exactive HF to yield more than ~1,600 protein quantifications across a 35-minute gradient. Being not as familiar with LFQ in MaxQuant as other quantification methods, it's unclear from the text how many unique peptide and protein identifications this corresponds to. In our experience, an instrument/sample combination such as the authors' should identify roughly 4K - 5K proteins under these conditions, which raises questions about losses during SP3 (automated or otherwise).

It is interesting that LFQ seems to normalize some of the variability of the authors' workflow. In Figure 4 for example, the number of LFQ protein groups scales very nicely with input - but the sum of the LFQ intensities does not (see e.g. 156ng and 100cell samples). I guess it's possible that some proteins exhibit dynamic recoveries during SP3, or that losses for some proteins are absolute at or below a certain threshold of abundance. In any case, some discussion of this phenomenon is warranted, perhaps in the context how LFQ functions.

The analysis of human ADC tumor samples is a useful example of the approach.

In summary, this is a useful approach that is expected to complement existing methods and add significantly to the toolbox of sample handling procedures for the field. A few minor tweaks will hopefully serve to further elevate an already nice body of work.

Reviewer #2:

Muller et al describe the automation of their previously described SP3 protocol on the Bravo liquid handling system from Agilent. They performed a detailed analysis of the reproducibility of the sample preparation by comparing intra- and inter-day variability and investigated the sensitivity by reducing the sample amount to only few ng starting material. To show the applicability of the automated workflow to clinical sample analysis, they followed with a study of NSCLC FFPE tumor sections. They analyzed the differences between regions with distinct histopathological subtype within single tumors, and characterized the functional associations of these differences. The entire study is based on the SP3 method, developed by the same lab, and already successfully implemented

by many other labs in the field. The application of the method to FFPE was also shown before by Hughes (previously in the Krijgsveld lab). Overall, this is a nicely presented study, and the analyses are very solid. However, despite the importance of automation, I think it is a rather incremental technological development. Nevertheless, the additional Bravo protocols will surely be useful to the community. There are several points that still require some attention:

1. The entire manuscript is focusing on the reproducibility of the analysis, and shows the CVs of commonly identified proteins. It is critical to assess the reproducibility of the identification rate, how many missing values, the association of missing values with intensity etc. They refer to missing values only in a single sentence (p26), which is insufficient to evaluate the robustness of the analyses.
2. The ability to recover proteins from very low amounts of input material is impressive. However, they indicate the starting amount and the number of identified proteins, which largely depends on the LC-MS parameters/performance. It would be more informative to know the exact recovery from each amount of starting material. They should quantify the amount of peptides before loading on the HPLC (at least for the larger amounts that are within the detection range of common quantification methods).
3. The clinical sample analyses show higher levels of ECM proteins in the lepidic samples. Similar protein families are also shown for the papillary subtype. The string network shows also higher levels of stromal components, such as immune cells, IFN-related proteins, neutrophil markers etc. These results suggest that the tissue dissection didn't sample the actual cancer cells, but more of the stroma. It is therefore not convincing that these differences represent the biology of these subtypes, and not a sampling artifact. In order to have better assessment of the biological relevance of their findings, they can try to correct for tumor sample cellularity. A better approach could analyze the histology. I understand that the manuscript is focused on the technology and not tumor biology and therefore understand if these analyses are beyond the scope of this study. Nevertheless, the authors should at least refer to these issues in writing.
4. In the analysis of the clinical samples, the authors define significantly changing proteins as having $p\text{-value} < 0.05$ (methods and figure). I assume that they couldn't find truly significant proteins (with FDR correction), but if so, they should not refer to those as significant.
5. It is not clear how the enrichment analysis was performed and what background was used. The String database usually uses the entire genome/proteome as background, which is not the appropriate background when the coverage is rather low with clear biases against lowly expressed proteins. Instead, they should use the entire identified dataset. In addition, the presentation of the enrichment analyses in the Supplementary Figures 9 and 10 is not clear. Is each category only significantly enriched in one direction? Are there categories that are de-enriched? Maybe a different visualization would be clearer.
6. Supp Figure 3 indicates #LFQ intensities, should be corrected to #LFQ proteins.
7. Tables at the end of the submitted pdf should be removed.

Reviewer #3:

Reviewer comments on the manuscript `Automated sample preparation with SP3 for low-input clinical proteomics`

Summary

Müller et al. describe in their manuscript entitled `Automated sample preparation with SP3 for low-input clinical proteomics` the automation of the SP3 protocol on an Agilent Bravo robotic system that is suitable for high-throughput applications. The aim of the manuscript is to supply a robotic solution for clinical samples with limited input material. After describing the platform, the authors evaluated the workflow with intra-day and inter-day tests to assess repeatability and reproducibility, which they report by calculating the coefficient of variation (CV) of quantified proteins. They applied the workflow to tissue lysates of macro-dissected tumor samples and dilution series of HeLa cells and HeLa cell lysates to illustrate that low amounts of input material can be processed with their pipeline.

General remarks

Jeroen Krijgsveld's laboratory invented the SP3 protocol several years ago (Hughes et al., MSB 2014) and the protocol itself is state-of-the-art and easy to use, which resulted in broad adaption in the proteomics community. Moreover, the mechanism of this protocol, which is based on protein

aggregation on sub-micron particles, has been described recently (Batth et al, MCP, 2019), making the protocol even more trustworthy. The manuscript describes the automation of parts of this protocol and the authors did an extensive evaluation of its analytical performance with intra-day variation, inter-day variation analyses and cross-contamination tests. The availability of the software and the fact that the Agilent Bravo systems are widely spread in laboratories can result in a broad adaption of this robotic workflow.

The application to minimal sample amounts will be interesting for readers in the proteomics community in which several labs are in a race towards single cell analysis. The tumor tissue as a low input material is a valid example, but tissue slices of 5 mm x 5 mm x 5 μ m are still far away from single cell analysis and can also be handled by other protocols. Nevertheless, the combination of low sample amounts and automation gives the manuscript an interesting technical aspect, which will be the largest advance of this manuscript.

Having said that the work in the manuscript was carried out well, it still lacks novelty and suffers from the fact that the workflow is only partially automated. Even though automation of sample preparation is fantastic to guarantee reproducibility in cases where large-numbers of samples are necessary to process, automation of this part of the proteomics workflow is not novel itself and has been even realized over the complete sample preparation workflow and not only starting from lysed samples. This has been facilitated before by several scientific laboratories and several companies like PreOmics and Agilent - both having a product pipeline running on the automation of the proteomic sample preparation (more information below).

The performed clinical study within the manuscript has a medium to large size for a proteomics study analyzing tissue samples and is interesting to read. The study and its analysis are performed well.

The manuscript itself reads very nicely and the figures are well drafted.

Overall, the authors present an interesting protocol for the partly automation of the previously described SP3 protocol, which lacks novelty and the automation is overstated. To make the protocol truly interesting for the community, the complete workflow from the raw sample to the ready-to-inject peptides would have to be automated without any manual interference.

Major points

1. Automation of the sample preparation step in MS-based proteomics is crucial to guarantee reproducibility for the processing of high numbers of samples. However, automation of this part of the proteomics workflow is not novel itself. This has been described and implemented in previous studies, even on the Agilent Bravo system (page 6, second paragraph). Moreover, Agilent has application notes for sample preparation workflows available and supplies the software protocol files ready-to-use (<https://www.agilent.com/cs/library/applications/5991-3602EN.pdf>; <https://www.agilent.com/cs/library/applications/application-protein-sample-preparation-preomics-bravo-proteomics-5994-0306en-agilent.pdf>). Even the complete automation of the sample preparation for various tissues is already possible. For example PreOmics sells an instrument for the automated sample preparation starting from cells/tissues to ready-to-inject samples (<https://www.preon.preomics.com/>). Furthermore, the potential to automate the SP3 protocol has been suggested before (Batth et al., MCP, 2019). Consequently, the work in this manuscript is not novel in terms of supplying an automated platform for clinical samples.

2. Key sample preparation steps requiring intensive hands-on time such as the processing until the lysate is generated from a cell line or a tissue sample are not automated in the manuscript. This includes sample solubilization, boiling and sonication to enable efficient protein extraction. These steps are crucial and easily subjected to proteomic variation and should be integrated in the automated pipeline.

The extensive time period for pre-heating is a valid reason for having an external device, but this only leaves the extraction of proteins from the lysate and the pipetting of enzymes as automated steps. In this manuscript more steps are manual than automated and questions the 'automated sample preparation' in the title of the manuscript. Moreover, it seems that the authors describe further hands-on steps needed for recovering and acidification of the peptides: "Following enzymatic digestion, peptide samples can be manually recovered and acidified in new plates or tubes (Figure 1G), or this can optionally be done on-deck after supplying new pipette tips". In summary, only the aliquoting of beads and the organic solvents buffer exchange during the washing steps (which is specific to this protocol) are automated. All other critical steps, including sample lysis, digestion and harvesting of the peptides are still done manually. This would strongly limit the adaption of this protocol by other laboratories.

The authors claim at several passages of the manuscript that the protocol eliminates hands-on time,

which should be rectified. Eliminating of hands-on time would mean that raw samples are placed in the robot and ready-to-inject peptides are harvested afterwards without any interference from the user. This is also the impression that the authors give the reader.

3. The experiments for assessing the intra-day and inter-day variation are valid for this propose in general but still have some experimental and analytical issues. The assessment of the CV's should be discussed in the manuscript. Clinical assays and platforms that might be suited for this process in future have to be assessed with all transparency. The protocol in the manuscript starts with samples that were already in a very advanced state of the sample preparation pipeline, which prevents the calculation of the repeatability and reproducibility of the workflow. All these steps will contribute to further variation. The authors aim to have an assessment according to clinical guidelines, but such an assessment have to include all parts of the sample preparation workflow and not only a sub process. This fact should be openly stated and discussed in the manuscript.

4. Focusing on proteins without any missing values is cherry-picking for the calculation of CVs. These proteins will be the highest abundant and most reproducible quantified proteins. This is also reflected by only looking at the 1650-1700 proteins from the probably 5000(?) quantified proteins in this analysis. The mass spectrometer was operated in DDA mode, selecting peptides in a semi-stochastic fashion for sequencing and missing values are parts of our research. Nevertheless, most proteins will be present at several time points and CVs can be calculated and they should not be excluded from the analysis. The transparent way for reporting CVs would mean to include all proteins with at least three values.

5. In samples with limited amounts of starting material the match-between-run algorithm tends to assign peptide `identifications` to background ions. Did the authors apply the match-between-run algorithm for the low sample amounts? If this was the case, the numbers without matching should be reported and correlations to measurements with higher sample amounts should be made to supply convincing data for correct peptide assignments.

6. The SP3 protocol is a fantastic proteomics sample preparation protocol. Nevertheless, throughout the manuscript other protocols are described with negative attributes that are not true.

Statements like "Among the most popular sample preparation methods, stage tips, and its derivative iST, do not tolerate detergents thereby restricting their generic use." or "The ability to handle detergent-containing samples, including SDS, is a great benefit over other methods such as iST, ..." (page 36) do other state-of-the-art proteomics sample preparation technology no justice. Samples from all kinds of tissues and plant materials were processed with protocols like the iST or with the help of other solid phase-extraction protocols, resulting in the most extensive proteomes ever reported (Coon et al., *Nat. Biotech.*, 2016; Bekker-Jensen et al., *Cell Syst.*, 2017; Doll et al., *Nat. Comm.*, 2017). Strong detergents like SDS are not necessary in general and in cases where someone wants to use them, they can be removed by protein precipitation.

Minor points

- The cited guideline for the evaluation of an analytical platform is thought to be for "veterinary drug residue studies". In general, it might be more appropriate to cite the FDA CLIA guidelines or recommendations from the proteomics community that are based on them (Grant and Hoofnagle, *Clin Chem*, 2014).

- The timelines for the analysis as they are depicted in supplemental Fig. 1 are very interesting as they give the reader an instant understanding of the protocols and could be shown already in the main Fig. 1

- Fig. 2D: Can the authors comment on the relative poor correlation of the manual SP3 and the robotic protocol? What could be the reason for this effect? This might be important to report as this/these factor/s might have an effect on the sample preparation itself and could help to further optimize the protocol.

- Page 28, paragraph 2: Please state the size of the bins e. g. 1000 proteins, 2 orders of magnitude for each bin

- The authors claim that the sample preparation only takes 1.5h (page 25, page 37), allowing the capacity of hundreds of samples. This is an overstatement and should be avoided as it leads to a misinterpretation for the reader (see above major point 2). The 1.5h includes only the purification of the proteins. Other sample preparation steps are neglected and they prevent that hundreds of samples can be prepared. The samples have to be lysed, RNA/DNA removed, disulfide bridges have to be reduced and alkylated (taking 45-65 minutes according to the authors) and most importantly the proteins have to be digested (2-16h; page 23). To avoid a misunderstanding the authors should additionally name the time for the complete sample preparation, when describing the whole sample preparation time and correct the potential of how many samples can be prepared.

- The reviewer does not agree that "filtration, centrifugation, precipitation and electrophoresis are difficult to standardize and unsuitable for automation". Especially the first two steps are already implemented in almost all robotic platforms like the Tecan, Hamilton or Agilent systems.
- In the method section it would be useful to know which magnetic rack was used in the Bravo system.
- Some references e.g. 26, 38, 39 start with first names. Please review all references.
- If a lid was necessary, how did the authors ensure that no droplets from condensate cross-contaminated the samples. Did the authors apply additional centrifugation steps e.g. to ensure that no condensate was on the lid or that air bubbles were removed from the buffers? If so, it would be helpful to inform the readers about this steps and supply advises how to avoid contamination e.g. by centrifugation and supply information about the lid. If lids with glue or heat-seal were tested, it would be good to report the details on the lids that were used in the material section and comment on potential polymer contamination in the mass spectra.

Typos:

- Page 12, line six: The degree {degree sign} symbol is missing for 95{degree sign}C
- Page 25, second paragraph, third line: EMA \diamond EMEA
- Page 32: Hematoxylin and Eosin is written with a different font than the rest of the text and 'Hematoxylin` is written in lower case and 'Eosin` in upper case
- Figure 5: The annotation of the different tumor regions is written in capitals in A and lower case in B and D

Authors' appeal

3rd September 2019

Thank you for sending the reviewer reports, we are grateful to the reviewers for devoting their time to carefully assess our manuscript. Yet, we are slightly disappointed by its rejection at this point, especially since all 3 see the value of the approach as an 'incredibly useful' development for the field. In addition, we feel that the reasons for the negative judgement are not completely fair, especially by reviewer 3: (s)he over-credits existing methods for their ability to start from raw samples (which they cannot), while dis-crediting automated SP3 for its supposed inability to start from such samples (which it can). The latter was a bit hidden in the manuscript, not describing this in detail for HeLa samples, which we'll be happy to elaborate upon if given the chance to send a revised manuscript. In addition, we already have proof-of-principle data in the lab that autoSP3 can be seamlessly combined with automated (96-well) sonication for tissue lysis and protein extraction, eliminating all manual sample handling. We have started to repeat and extend these experiments in a more consistent manner for various organs, and will be happy to include this in a revised manuscript.

I am attaching a point-to-point response to all raised issues, including the main concerns alluded to above, i.e. explaining the limitations of previous methods with regard to the type of starting material (lysate, not tissue), and emphasizing the steps included in automated SP3 (including reduction/alkylation prior to the core SP3 protocol). In addition, we explain the experiments that we are currently performing to get from tissues to MS-ready peptides without any manual handling. This can be completed in the very near future. Furthermore, we respond to all other raised issues, we include revised figures, and indicate how text will be changed in a revised manuscript. Collectively, we are convinced that this represents an extremely attractive workflow for any proteomics lab that will spur a lot of innovative work. Speaking about this, but more as a side note, it occurred to me that the method presented in the Villen-paper starts from yeast lysates, which thus could be equally criticized for lack of sufficient innovation.

Taking this all together, I hope that you agree with the proposed path forward, and that you are willing to give us the opportunity to send a revised manuscript.

Reviewer #1:

In this manuscript from Mueller and colleagues (hereafter referred to as the Mueller Report), the authors advance their solid-phase protein extraction and digestion

method to an automated liquid-handling robot and validate its performance. This "automated SP3" sample preparation is then challenged with low sample inputs (protein from ~100 cells) and a real-world application in profiling protein abundance differences from human patient-derived FFPE samples of lung adenocarcinoma tumors of various histological subtypes.

This is a timely, informative and useful study that is sure to help researchers in the field of quantitative proteomics work more efficiently, reproducibly and easily on their research projects. It's pretty well written, although I have a few minor suggestions to streamline the text a bit. Although I hate to bring up the dreaded "conceptual innovation" topic, it should be clear that while incredibly useful and artfully described and validated, this is not a novel or new idea implemented for the first time. In my view, however, the utility of these methods far outweighs the often overwrought emphasis on novelty.

Response: We thank the reviewer for his/her positive assessment, seeing the potential of automated SP3 to enhance reproducible data generation in proteomics.

To enhance innovation, **we have included new data** in the revised manuscript showing the processing of raw tissue samples all the way to peptides eliminating all manual handling steps, i.e. now including an automated tissue lysis step that seamlessly interfaces with automated SP3. Specifically, we collected 3 types of tissue (liver, kidney, heart) that were lysed in an automated and highly controlled manner using a Covaris sonicator, enabling processing of 96 samples simultaneously. These samples were next transferred to the Bravo robot for automated reduction/alkylation and SP3 all the way to peptides, directly followed by LCMS analysis (see new figure X). Importantly, we used <1mg wet tissue as an input, demonstrating that the method is fully compatible with low-input applications. **Collectively, this demonstrates that tissue-proteomics can be performed in an automated manner without manual handling steps.**

There is considerable effort taken in the early part of the manuscript to describe things that were attempted to be automated but failed. This reads a little too conversational; if space considerations are important, much of this could be streamlined. A more succinct depiction of the workflow and the automation in the early Results section might be better served in the main text, with anecdotes reserved for maybe Supporting Information, if at all.

Response: We can see the reviewer's point, and we have made changes as suggested to achieve a more succinct description of the workflow. However, we have kept some of the description as we think it is important to understand the details of the protocol. In addition, implementation of some handling steps is not completely trivial, requiring special care when transferring the manual protocol to a robotic system. We believe that pointing to this will be useful for potential troubleshooting or for execution of SP3 on other liquid handling platforms.

To readers not familiar with the initial SP3 publication, comparison of an automated with manual SP3 method may not seem rigorous enough. Some additional discussion (or a simple comparison experiment) on traditional FASP/8M urea/etc. preparation and digestion approaches might be helpful here (even self-citations to the 2014 paper). For example, one might expect 1ug of HeLa cell digest analyzed on a Q-Exactive HF to yield more than ~1,600 protein quantifications across a 35-minute gradient. Being not as familiar with LFQ in MaxQuant as other quantification methods, it's unclear from the text how many unique peptide and

protein identifications this corresponds to. In our experience, an instrument/sample combination such as the authors' should identify roughly 4K - 5K proteins under these conditions, which raises questions about losses during SP3 (automated or otherwise).

Response: As this is not a primary method development or method comparison article, we have not detailed all proof-of-concept experiments regarding the performance of SP3, which we investigated in more detail in our earlier work (Hughes et al., MSB 2014, Hughes et al., Nat Protocols 2019). In addition, SP3 has been compared to other widely accepted methods (e.g. FASP, iST) by several independent groups (e.g. Sielaff et al., JPR 2017, Dagley et al., JPR 2019). However, we have tried to clarify this in the text by citing published research that demonstrates the performance and benefits of SP3 compared to other methods.

We assume that the second part of the reviewer's comment refers to the results shown in Figure 2C. Here, we focused on proteins that have been identified and quantified without missing values across each individual day (n=10 per day). We like to point out that this is only a subset of the total number of identified proteins. The overall number of quantified proteins per sample is provided in Supplementary Figure 3 with an average of 3191 proteins. The data are in range with our quality control (QC) for the setup. Although we agree that higher identification rates may be possible with other chromatographic set ups, this will not affect the performance metrics of SP3. Since Figure 2C raised confusion for all 3 reviewers, we have revised it to enhance clarity, now including all data with at least 3 valid LFQ values across all 60 data sets (see below). All CVs remain <15%, hence our conclusions remain unaffected.

With regard to potential protein loss during SP3, we like to point the reviewer's attention to Figure 4A. Here, we injected the equivalent of 1 microgram protein from the four highest starting materials (1.25 μ g, 2.5 μ g, 5 μ g, and 10 μ g). The resulting plateau of equal numbers of quantified proteins and summed intensities indicates a comparable recovery and also shows a range of ~2000 quantified proteins from roughly 1 μ g, indicating efficient protein capture and recovery even for the highest sample loads (10 μ g). In addition, we have compared recovery by manual and automated SP3 for a 10 μ g HeLa sample, indicating very similar performance (or even slightly better recovery for automated SP3) (Review Figure 1, below).

Revised Figure 2C, to contain all proteins with at least 3 valid values across 60 samples.

Review Figure 1: Comparison of ion intensities in LCMS after processing a HeLa lysate by automated SP3 (top panel) and manual SP3 (bottom).

It is interesting that LFQ seems to normalize some of the variability of the authors' workflow. In Figure 4 for example, the number of LFQ protein groups scales very nicely with input - but the sum of the LFQ intensities does not (see e.g. 156ng and 100cell samples). I guess it's possible that some proteins exhibit dynamic recoveries during SP3, or that losses for some proteins are absolute at or below a certain threshold of abundance. In any case, some discussion of this phenomenon is warranted, perhaps in the context how LFQ functions.

Response: We agree and we thank the reviewer for pointing this out: indeed LFQ includes a normalization step which is not intended for quantifying proteins from sample loadings with orders of magnitude difference. The algorithm assumes that the total protein amount within each sample is identical or at least similar. Here, this is of course not the case and we have updated Figure 4 and corresponding text according to the reviewer's suggestion. Specifically, we changed LFQ to intensity-based absolute quantification (iBAQ) values to avoid any normalization across samples, now producing the expected trend (see revised figure 4 below).

Based on our own experience (Hughes et al., MSB 2014, Hughes et al., Nat Protocols 2019) and the studies of other independent groups (e.g. Sielaff et al., JPR 2017, Dagley et al., JPR 2019) there is no indication of dynamic recovery of particular proteins, which is one of the hallmarks of SP3.

Revised Figure 4, using iBAQ instead of LFQ values for assessment of SP3-performance.

The analysis of human ADC tumor samples is a useful example of the approach. In summary, this is a useful approach that is expected to complement existing methods and add significantly to the toolbox of sample handling procedures for the field. A few minor tweaks will hopefully serve to further elevate an already nice body of work.

We thank the reviewer for the useful comments to improve our manuscript.

Reviewer #2:

Muller et al describe the automation of their previously described SP3 protocol on the Bravo liquid handling system from Agilent. They performed a detailed analysis of the reproducibility of the sample preparation by comparing intra- and inter-day variability and investigated the sensitivity by reducing the sample amount to only few ng starting material. To show the applicability of the automated workflow to clinical sample analysis, they followed with a study of NSCLC FFPE tumor sections. They analyzed the differences between regions with distinct histopathological subtype within single tumors, and characterized the functional associations of these differences. The entire study is based on the SP3 method, developed by the same lab, and already successfully implemented by many other labs in the field. The application of the method to FFPE was also shown before by Hughes (previously in the Krijgsveld lab). Overall, this is a nicely presented study, and the analyses are very solid. However, despite the importance of automation, I think the it is a rather incremental technological development. Nevertheless, the

additional Bravo protocols will surely be useful to the community. There are several points that still require some attention:

1. The entire manuscript is focusing on the reproducibility of the analysis, and shows the CVs of commonly identified proteins. It is critical to assess the reproducibility of the identification rate, how many missing values, the association of missing values with intensity etc. They refer to missing values only in a single sentence (p26), which is insufficient to evaluate the robustness of the analyses.

Response: We believe that there must have been a misunderstanding since commonly identified proteins were solely considered in Figure 2C, which illustrates the CVs of commonly identified and quantified proteins within each day. Here, the relative differences in the comparison of median and average CV's between days is relevant rather than the absolute levels of CV percentages. However, since Figure 2C raised confusion for all 3 reviewers, we have revised it to enhance clarity, now including all data with at least 3 valid LFQ values across all 60 data sets (see Revised Figure 2C below). All CVs remain <15%, hence our conclusions remain unaffected.

In all subsequent analyses, we focused on proteins with an LFQ intensity in at least 45 out of 60 raw files (75% data completeness), covering 79.04% of the entire list of proteins in the dataset. In addition, we reported the median and average CV's observed from the complete list of quantified proteins (n=3750) across all 60 samples to be 18.1% and 20.5%. Again, to avoid further confusion, we extended Table 1 to include CV's for the complete list of proteins with minimum three valid values across all samples (see Revised Table 1 below).

We agree with the reviewer that, beyond CVs, identification rates and missing values are important parameters for proteomics experiments. These data are in fact provided in the manuscript, although admittedly a bit hidden. For instance, the identification rate per sample is provided in Supplementary Figure 3. We had not included data on the association of missing values with intensity, since this follows the expected trend that proteins of low abundance are identified with a higher number of missing values (see Revision Figure 2 below). This is typical for any shotgun proteomics experiment and does not inform on the reproducibility of the experiment, and therefore showing Revision Figure 2 in the manuscript will not provide insight in the performance of autoSP3. Instead, the effect of protein abundance (and thereby indirectly of the number of missing values) on CV follows from Figure 3A and Supplementary Figure 5.

Revised Figure 2C, to contain all proteins with at least 3 valid values across 60 samples

Table 1

LFQ Intra- & Inter-day Variability	# of replicates	75% Data Completeness (n=2964)		Minimum of 3 valid values (n=3688)	
		Median CV [%]	Average CV [%]	Median CV [%]	Average CV [%]
within Day -1	10	11.9	14.6	13.3	16.5
within Day -13	10	9.8	12.2	11.3	14
within Day -27	10	10.8	13.5	12.3	15.5
across Days	30	13.3	15.7	16	18.6
w/o MS variability	30	14.3	17.3	17.2	20.1
Overall automated	60	14.7	17.4	18.1	20.6
Manual SP3	16	16.3	18.6	17.3	20

Revised Table 1, extended to now report CVs of proteins with a minimum of 3 valid values.

Revision Figure 2. Correlation between protein intensities and missing values.

2. The ability to recover proteins from very low amounts of input material is impressive. However, they indicate the starting amount and the number of identified proteins, which largely depends on the LC-MS parameters/performance. It would be more informative to know the exact recovery from each amount of starting material. They should quantify the amount of peptides before loading on the HPLC (at least for the larger amounts that are within the detection range of common quantification methods).

Response: We agree with the reviewer that a quantification of peptide amounts prior to the injection would be informative. However, we would like to note two points: i) as the reviewer already indicated, for the majority of samples no common quantification method (peptide assay) would be sufficiently sensitive to generate convincing data. ii) it is nearly impossible to compare a sample before and after SP3 by MS. In particular, it would require the removal of SDS from the sample not treated by SP3 by an alternative method. Even if this can be achieved, the comparison would be between SP3 and the alternative method, not between a sample before and after SP3.

If the main interest or concern of the reviewer is the general recovery of peptides from a given protein input during the SP3 protocol, this has been extensively validated in our previous work (Hughes et al., MSB 2014, Hughes et al., Nat Protocols 2019) and by other independent groups (e.g. Sielaff et al., JPR 2017, Dagley et al., JPR 2019). To indicate to the reviewer that performance of automated SP3 is equivalent to manual SP3, we have compared recovery by manual and automated SP3 for a 10 μ g HeLa sample, indicating very similar performance (or even slightly better recovery for automated SP3) (Review Figure 1, below).

Review Figure 1: Comparison of ion intensities in LCMS after processing a HeLa lysate by automated SP3 (top panel) and manual SP3 (bottom).

3. The clinical sample analyses show higher levels of ECM proteins in the lepidic samples. Similar protein families are also shown for the papillary subtype. The string network shows also higher levels of stromal components, such as immune cells, IFN-related proteins, neutrophil markers etc. These results suggest that the tissue dissection didn't sample the actual cancer cells, but more of the stroma. It is therefore not convincing that these differences represent the biology of these subtypes, and not a sampling artifact. In order to have better assessment of the biological relevance of their findings, they can try to correct for tumor sample cellularity. A better approach could analyze the histology. I understand that the manuscript is focused on the technology and not tumor biology and therefore understand if these analyses are beyond the scope of this study. Nevertheless, the authors should at least refer to these issues in writing.

Response: We agree with the reviewer's concern about tumor cell content (TCC). We therefore added a new figure 5E complementary to Figure 5B, now color-coded for TCC in each sample (see below). Since TCC distribution is random, this indicates that the separation is not driven by the TCC. One might argue that the separation of papillary_1 and _2 subgroups may be the exception, although their average and median TCC is rather similar. We will point out the potential influence of TCC in the manuscript when discussing this new figure 5B.

New Figure 5E. tSNE analysis of proteome profiles for 51 adenocarcinoma samples, color coded for tumor content

4. In the analysis of the clinical samples, the authors define significantly changing proteins as having p -value < 0.05 (methods and figure). I assume that they couldn't find truly significant proteins (with FDR correction), but if so, they should not refer to those as significant.

Response: we agree that we should have included the p -value correction from the beginning. We updated the corresponding parts of the manuscript (Text, Methods, and Figures).

5. It is not clear how the enrichment analysis was performed and what background was used. The String database usually uses the entire genome/proteome as background, which is not the appropriate background when the coverage is rather low with clear biases against lowly expressed proteins. Instead, they should use the entire identified dataset. In addition, the presentation of the enrichment analyses in the Supplementary Figures 9 and 10 is not clear. Is each category only significantly enriched in one direction? Are there categories that are de-enriched? Maybe a different visualization would be clearer.

Response: We added clarification regarding the STRING-based gene ontology enrichment analyses. In brief, we used the entire dataset as a specific background consisting of 5642 proteins. In addition, we required highest confidence (minimal required interaction score of 0.9) and excluded text mining-based evidence. Disconnected nodes in the network analyses were hidden. All remaining settings were left at default.

6. Supplementary Figure 3 indicates #LFQ intensities, should be corrected to #LFQ proteins.

Response: this has now been corrected.

7. Tables at the end of the submitted pdf should be removed.

Response: we will clarify this with the handling editor and adhere to MSB guidelines.

Reviewer #3:

Reviewer comments on the manuscript `Automated sample preparation with SP3 for low-input clinical proteomics`

General remarks

Jeroen Krijgsveld's laboratory invented the SP3 protocol several years ago (Hughes et al., MSB 2014) and the protocol itself is state-of-the-art and easy to use, which resulted in broad adaption in the proteomics community. Moreover, the mechanism of this protocol, which is based on protein aggregation on sub-micron particles, has been described recently (Batth et al, MCP, 2019), making the protocol even more trustworthy. The manuscript describes the automation of parts of this protocol and the authors did an extensive evaluation of its analytical performance with intra-day variation, inter-day variation analyses and cross-contamination tests. The availability of the software and the fact that the Agilent Bravo systems are widely spread in laboratories can result in a broad adaption of this robotic workflow.

The application to minimal sample amounts will be interesting for readers in the proteomics community in which several labs are in a race towards single cell analysis. The tumor tissue as a low input material is a valid example, but tissue slices of 5 mm x 5 mm x 5 µm are still far away from single cell analysis and can also be handled by other protocols. Nevertheless, the combination of low sample amounts and automation gives the manuscript an interesting technical aspect, which will be the largest advance of this manuscript.

Having said that the work in the manuscript was carried out well, it still lacks novelty and suffers from the fact that the workflow is only partially automated. Even though automation of sample preparation is fantastic to guarantee reproducibility in cases where large-numbers of samples are necessary to process, automation of this part of the proteomics workflow is not novel itself and has been even realized over the complete sample preparation workflow and not only starting from lysed samples. This has been facilitated before by several scientific laboratories and several companies like PreOmics and Agilent - both having a product pipeline running on the automation of the proteomic sample preparation (more information below).

The performed clinical study within the manuscript has a medium to large size for a proteomics study analyzing tissue samples and is interesting to read. The study and its analysis are performed well. The manuscript itself reads very nicely and the figures are well drafted. Overall, the authors present an interesting protocol for the partly automation of the previously described SP3 protocol, which **lacks novelty** and the **automation is overstated**. To make the protocol truly interesting for the community, the **complete workflow from the raw sample to the ready-to-inject peptides would have to be automated** without any manual interference.

Response: We thank the reviewer for his/her overall positive assessment of the manuscript, and for seeing the value of automated SP3 for the proteomics community. Yet, we disagree with the major criticisms with regard to lack of novelty as well as overstatement of automation. As explained in the detailed comments below, this is most likely due to a misunderstanding, leading to an underestimation of what is included in automated SP3, and in overestimating what other methods do.

Major points

1. Automation of the sample preparation step in MS-based proteomics is crucial to guarantee reproducibility for the processing of high numbers of samples. However, automation of this part of the proteomics workflow is not novel itself. This has been described and implemented in previous studies, even on the Agilent Bravo system (page 6, second paragraph). Moreover, Agilent has application notes for sample preparation workflows available and supplies the software protocol files ready-to-use (<https://www.agilent.com/cs/library/applications/5991-3602EN.pdf>; <https://www.agilent.com/cs/library/applications/application-protein-sample-preparation-preomics-bravo-proteomics-5994-0306en-agilent.pdf>). Even the complete automation of the sample preparation for various tissues is already possible. For example PreOmics sells an instrument for the automated sample preparation starting from cells/tissues to ready-to-inject samples (<https://www.preon.preomics.com/>). Furthermore, the potential to automate the SP3 protocol has been suggested before (Batth et al., MCP, 2019). Consequently, the work in this manuscript is not novel in terms of supplying an automated platform for clinical samples.

Response: We agree that several parts of the proteomic workflow have been implemented on different robotic platforms, using various methods. The iST method, used both in the Agilent and Preomics platforms cited above, is one of them, however these procedures are neither generic nor fully automated as suggested by the reviewer, thereby contrasting to automated SP3 for various reasons:

- i) Solid-phase extraction and reversed phase chromatography, like iST, are incompatible with the use of SDS or other detergents (e.g. as is preferred for FFPE tissue, see below), thereby limiting the use of both the Agilent and Preomics protocols. In fact, the cited protocol (<https://www.agilent.com/cs/library/applications/application-protein-sample-preparation-preomics-bravo-proteomics-5994-0306en-agilent.pdf>) is a method to process plasma samples where no protein extraction is needed. We even mention in the main text: “Automated sample preparation is far less common in proteomics, and is restricted to cases where detergents can be avoided (iST, plasma proteomics21)”. We believe that the ability to use SDS is a very strong asset of SP3 especially for FFPE tissues: heating in SDS is the preferred way for FFPE protein extraction (e.g. Wisniewski, Dus and Mann, Prot Clin Appl 2013), pre-empting the use of iST including the protocols cited by the reviewer. This is further illustrated by the fact that the original iST paper (Kulak et al, 2014) has been cited 674 times, however none of these studies used it for FFPE tissue (based on Google Scholar: an ‘in-article’ text search for the term FFPE retrieves 15 out of 674 papers. Detailed reading these 15 papers indicated that none used iST). Therefore, autoSP3 fills an important need.
- ii) **Neither of the cited protocols include a tissue extraction step:** they both start from (detergent-free) samples. Therefore, contrary

to the reviewer's claim, the **Preomics protocol does NOT start from tissues.**

- iii) The cited Agilent protocol includes several centrifugation steps which are performed OFF the deck of the Bravo system. This contrasts with autoSP3 which is completely performed ON the deck (i.e. no manual intervention).
- iv) The Preomics instrument can process 12 samples simultaneously, compared to 96 for autoSP3.
- v) The existence of the Preomics instrument does not remove the novelty of our automated SP3 protocol, since it adds the key benefits of SP3 compared to iST as discussed in several independent studies (e.g. Sielaff et al., JPR 2017). In addition, the Bravo liquid handling platform is widely available in many genomics and biochemistry laboratories which facilitates adoption of the method by the proteomic community.

It is correct that automation of the SP3 protocol has been suggested before, however this was first done by ourselves when introducing SP3 (Hughes et al., MSB 2014) and not by Batth et al (2019), since this potential was immediately clear from the inception of SP3. Furthermore, we hope that the reviewer appreciates that there is a big difference between suggesting a method, and effectively realizing and implementing it. A method to process 96 samples in parallel for low-input proteomics did not exist before, hence this is an important innovation.

2. Key sample preparation steps requiring intensive hands-on time such as the processing until the lysate is generated from a cell line or a tissue sample are not automated in the manuscript. This includes sample solubilization, boiling and sonication to enable efficient protein extraction. These steps are crucial and easily subjected to proteomic variation and should be integrated in the automated pipeline.

The extensive time period for pre-heating is a valid reason for having an external device, but this only leaves the extraction of proteins from the lysate and the pipetting of enzymes as automated steps. In this manuscript more steps are manual than automated and questions the 'automated sample preparation' in the title of the manuscript. Moreover, it seems that the authors describe further hands-on steps needed for recovering and acidification of the peptides: "Following enzymatic digestion, peptide samples can be manually recovered and acidified in new plates or tubes (Figure 1G), or this can optionally be done on-deck after supplying new pipette tips". In summary, only the aliquoting of beads and the organic solvents buffer exchange during the washing steps (which is specific to this protocol) are automated. All other critical steps, including sample lysis, digestion and harvesting of the peptides are still done manually. This would strongly limit the adaption of this protocol by other laboratories. The authors claim at several passages of the manuscript that the protocol eliminates hands-on time, which should be rectified. Eliminating of hands-on time would mean that raw samples are placed in the robot and ready-to-inject peptides are harvested afterwards without any interference from the user. This is also the impression that the authors give the reader.

Response: These concerns must be based on a misunderstanding, possibly caused in part by the diverse of workflows that are described in the manuscript where steps can be performed on or off the liquid handling platform as preferred by the user. In addition, the reviewer likely underestimates the inherent simplicity of the SP3 workflow, meaning that there is a limited number of steps that need to be automated. For clarity: protocols A, B, C, and D (Supplementary Figure S1) include all steps for automated reduction/alkylation and SP3 in an unattended process starting from a lysed sample to peptides ready for MS (i.e. combining reduction and alkylation, protein clean-up, digestion, and acidification). Thereby, **all steps can be performed on the robot** (Supplementary Figure 1). The only interference is the sealing of the sample plate and transferring it to an incubator for digestion, and moving it back to the robot for acidification (<30 seconds hands-on time). Optionally (and this may have created confusion), the user can choose to perform reduction/alkylation and acidification manually to result in a slightly faster (but interrupted) procedure.

To further emphasise the capability to eliminate any manual handling step including lysis, not only for cultured cells but also for tissue, **we have included new data** in the manuscript where 3 types of tissue (liver, kidney, heart) were lysed in an automated and highly controlled manner using a Covaris sonicator enabling processing of 96 samples simultaneously. These samples were next transferred to the Bravo robot for automated reduction/alkylation and SP3 all the way to peptides, directly followed by LCMS analysis (see new figure X). Importantly, we used <1mg wet tissue as an input, demonstrating that the method is fully compatible with low-input applications. **This demonstrates that tissue-proteomics can be performed in an automated manner without manual handling steps.**

Thus, we argue that we do eliminate hands-on time which is largely facilitated by the properties of SP3. Furthermore, we disagree with the reviewer and think that the provided protocols and detailed description in this study will strongly support quick and easy adaption of our protocol in the field (as is already happening).

3. The experiments for assessing the intra-day and inter-day variation are valid for this propose in general but still have some experimental and analytical issues. The assessment of the CV's should be discussed in the manuscript. Clinical assays and platforms that might be suited for this process in future have to be assessed with all transparency. The protocol in the manuscript starts with samples that were already in a very advanced state of the sample preparation pipeline, which prevents the calculation of the repeatability and reproducibility of the workflow. All these steps will contribute to further variation. The authors aim to have an assessment according to clinical guidelines, but such an assessment have to include all parts of the sample preparation workflow and not only a sub process. This fact should be openly stated and discussed in the manuscript.

Response: We thank the reviewer for the remarks regarding the assessment of CV's. We clarified in the text that we are talking about the repeatability and reproducibility of the automated SP3 workflow.

The comparison of samples processed using manual versus autoSP3 clearly shows that the overall reproducibility is very high (Fig 2D). Considering the high throughput achieved using the automated the manual processing of 96 samples would easily take several hours of manual pipetting, which is prone to errors the longer it takes.

To clarify sample handling prior to SP3: in the case the cell dilution series (10-10,000 cells, Figure 4B), cells were serially dispensed in SDS-containing lysis buffer in a micro-titer plate, **and the plate (i.e. all samples simultaneously) were sonicated in a waterbath sonicator** before transfer to the Bravo robot. We realize that this was not properly described in the methods section, which we corrected in the revised manuscript and has been better articulated in the results section. We show with n=8 replicates (processed on two different plates) that our workflow is highly reproducible from beginning to end of the sample processing. The error bars indicate high consistency of the results despite processing of quantity-limited samples. Showing proteomic data for samples as small as 100 cells or less is rare in itself, doing this in an automated manner has not been done before.

To further emphasise the capability to eliminate any manual handling step including lysis, not only for cultured cells but also for tissue, **we have included new data** in the manuscript where 3 types of tissue (liver, kidney, heart) were lysed in an automated and highly controlled manner using a Covaris sonicator, enabling processing of 96 samples simultaneously. These samples were next transferred to the Bravo robot for automated reduction/alkylation and SP3 all the way to peptides, directly followed by LCMS analysis (see new figure X). Importantly, we used <1mg wet tissue as an input, demonstrating that the method is fully compatible with low-input applications. **Collectively, this demonstrates that tissue-proteomics can be performed in an automated manner without manual handling steps.**

4. Focusing on proteins without any missing values is cherry-picking for the calculation of CVs. These proteins will be the highest abundant and most reproducible quantified proteins. This is also reflected by only looking at the 1650-1700 proteins from the probably 5000(?) quantified proteins in this analysis. The mass spectrometer was operated in DDA mode, selecting peptides in a semi-stochastic fashion for sequencing and missing values are parts of our research. Nevertheless, most proteins will be present at several time points and CVs can be calculated and they should not be excluded from the analysis. The transparent way for reporting CVs would mean to include all proteins with at least three values.

Response: We believe that there must have been a misunderstanding since we considered commonly identified proteins **solely in Figure 2C**. Here, the relative differences in the comparison of median and average CV's between days is relevant rather than the absolute levels of CV percentages. However, to avoid further confusion we also made changes according to the reviewer's remark to include all proteins with at least three valid intensities across n=10 samples within a processing day (see revised Figure 2C below). This only marginally affected CVs, (all <0.15), hence our conclusions remain valid.

In all other analyses described in the manuscript, we focused on proteins with an LFQ intensity in at least 45 out of 60 raw files (75% data completeness, see Review Figure 2), covering 79.04% of the entire list of proteins in the dataset. In addition, we reported the median and average CV's observed from the complete list of quantified proteins with >3 valid values across all 60 samples to be 18.1% and 20.6%. Again, to avoid further confusion, we extended Table 1 (below) to include CV's for the complete list of proteins.

Revised Figure 2C, to contain all proteins with at least 3 valid values across 60 samples

Review Figure 2. Selection of proteins with an LFQ intensity in 45 out of 60 runs, including 79% of all identified proteins.

Table 1

LFQ Intra- & Inter-day Variability	# of replicates	75% Data Completeness (n=2964)		Minimum of 3 valid values (n=3688)	
		Median CV [%]	Average CV [%]	Median CV [%]	Average CV [%]
within Day -1	10	11.9	14.6	13.3	16.5
within Day -13	10	9.8	12.2	11.3	14
within Day -27	10	10.8	13.5	12.3	15.5
across Days	30	13.3	15.7	16	18.6
w/o MS variability	30	14.3	17.3	17.2	20.1
Overall automated	60	14.7	17.4	18.1	20.6
Manual SP3	16	16.3	18.6	17.3	20

Revised Table 1, extended to now report CVs of proteins with a minimum of 3 valid values.

5. In samples with limited amounts of starting material the match-between-run algorithm tends to assign peptide `identifications` to background ions. Did the authors apply the match-between-run algorithm for the low sample amounts? If this was the case, the numbers without matching should be reported and correlations to measurements with higher sample amounts should be made to supply convincing data for correct peptide assignments.

Response: In both experiments of quantity-limited input material (Figure 4A and 4B) the match between runs algorithm was **not** used for obvious reasons as indicated by the reviewer. We clarified this in the manuscript to avoid confusion.

6. The SP3 protocol is a fantastic proteomics sample preparation protocol. Nevertheless, throughout the manuscript other protocols are described with negative attributes that are not true. Statements like "Among the most popular sample preparation methods, stage tips, and its derivative iST, do not tolerate detergents thereby restricting their generic use." or `The ability to handle detergent-containing samples, including SDS, is a great benefit over other methods such as iST, ...` (page 36) do other state-of-the-art proteomics sample preparation technology no justice. Samples from all kinds of tissues and plant materials were processed with protocols like the iST or with the help of other solid phase-extraction protocols, resulting in the most extensive proteomes ever reported (Coon et al., Nat. Biotech., 2016; Bekker-Jensen et al., Cell Syst., 2017; Doll et al., Nat. Comm., 2017). Strong detergents like SDS are not necessary in general and in cases where someone wants to use them, they can be removed by protein precipitation.

Response: It was not our intention to discredit any of the other methods. All of the mentioned approaches have their pros and cons and have their specific reasons to be established in the proteomic community. However, our statement points to the key properties where we believe SP3 stands out, e.g. not requiring solid-phase-extraction protocols or protein precipitation. Of course, most of the previously mentioned protocols can be combined with other methods and integrated in a compatible way to generate extensive proteomes. We have revised the text to remove the impression of discrediting previous work, while explaining assets of SP3.

It is correct that SDS is not necessary for all sample types, although advantageous for many. The prime reason why researchers resort to detergent-free extraction methods is that elimination of detergents is cumbersome yet necessary (e.g. for MS). Having SP3, researchers are free to choose whether they employ detergents, chaotropes, or other conditions for sample preparation. SDS may be removed by precipitation, however we hope that the reviewer agrees that this is not a realistic option for low-input samples e.g. <1000 cells (Figure 4b).

Minor points

- The cited guideline for the evaluation of an analytical platform is thought to be for `veterinary drug residue studies`. In general, it might be more appropriate to cite the FDA CLIA guidelines or recommendations from the proteomics community that are based on them (Grant and Hoofnagle, Clin Chem, 2014).

We thank the reviewer for the useful comment and we have included the reference in the manuscript.

- The timelines for the analysis as they are depicted in supplemental Fig. 1 are very interesting as they give the reader an instant understanding of the protocols and could be shown already in the main Fig. 1

We thank the reviewer for the useful comment, and we have moved the panel into Figure 1.

- Fig. 2D: Can the authors comment on the relative poor correlation of the manual SP3 and the robotic protocol? What could be the reason for this effect? This might be important to report as this/these factor/s might have an effect on the sample preparation itself and could help to further optimize the protocol.

Response: We think that there was a misunderstanding by the reviewer, possibly caused by the colour scheme in the Figure spanning a very narrow window in the very high SD-range (1 – 0.94), in fact indicating very high correlation between the manual and robotic protocol. As we already stated in the main text, this indicates that SP3 is highly robust in both formats, where SD of 0.94 between both versions (compared to >0.97 within them) may reflect subtle differences e.g. in sample volumes. In addition, there were several month between the manual and the automated processing, likely reflected in LCMS performance differences (e.g. different/new trap and/or analytical column, MS performance). We have adapted the text to avoid mis-interpretation of the Figure.

- Page 28, paragraph 2: Please state the size of the bins e. g. 1000 proteins, 2 orders of magnitude for each bin

Response: As already highlighted within the corresponding Figure 3A, we clarified the size of each abundance bin in the text.

- The authors claim that the sample preparation only takes 1.5h (page 25, page 37), allowing the capacity of hundreds of samples. This is an overstatement and should be avoided as it leads to a misinterpretation for the reader (see above major point 2). The 1.5h includes only the purification of the proteins. Other sample preparation steps are neglected and they prevent that hundreds of samples can be prepared. The samples have to be lysed, RNA/DNA removed, disulfide bridges have to be reduced and alkylated (taking 45-65 minutes according to the authors) and most importantly the proteins have to be digested (2-16h; page 23). To avoid a misunderstanding the authors should additionally name the time for the complete sample preparation, when describing the whole sample preparation time and correct the potential of how many samples can be prepared.

Response: We agree that we should have been more careful making our point, and that the mentioned throughput refers to the actual SP3 processing while lysis and digestion time come on top. Yet, these latter steps are performed in parallel to the SP3 procedure (off the robot deck), enabling a realistic scenario where several hundreds of samples can be processed per day, as follows: Tissues can be lysed in a 96-well format using a Covaris sonicator, as illustrated by our new data, taking ~60 min for 96 samples in an automated, highly controlled manner. This produces tissue extracts that can be automatically processed for reduction/alkylation (65 minutes) and autoSP3 (1.5h) on the Bravo. Next, the plate is incubated (off-deck) for proteolysis, making the Bravo robot available for a next plate of lysed tissue. Thus, in theory, this provides the capacity to run four 96-well plates in a normal eight-hour working day by a single operator. The next day, protein digests from these 4

plates can be recovered on the Bravo using the peptide recovery protocol (protocol D, Supplementary Figure 1), only taking 7.5 minutes per plate, producing ready-to-inject peptides in a new 96-well plate that can be immediately transferred to an LC autosampler. Whether this capacity can be exploited will depend on the LCMS method, the number of compounds that need to be detected for the intended assay, and the number of available mass spectrometers. At any rate, SP3 is unlikely to be a bottleneck in high-throughput applications, or in fact it will resolve a current bottleneck and move it to the next stage (i.e. MS analysis).

- The reviewer does not agree that "filtration, centrifugation, precipitation and electrophoresis are difficult to standardize and unsuitable for automation". Especially the first two steps are already implemented in almost all robotic platforms like the Tecan, Hamilton or Agilent systems.

Response: We agree with the reviewer and have made changes in the manuscript to clarify our statement. While filtration and centrifugation steps can be automated they require the appropriate instrumentation.

- In the method section it would be useful to know which magnetic rack was used in the Bravo system.

Response: We thank the reviewer for pointing to this omission, we have now added the information to the method section.

- Some references e.g. 26, 38, 39 start with first names. Please review all references.

This has now been corrected.

- If a lid was necessary, how did the authors ensure that no droplets from condensate cross-contaminated the samples. Did the authors apply additional centrifugation steps e.g. to ensure that no condensate was on the lid or that air bubbles were removed from the buffers? If so, it would be helpful to inform the readers about this steps and supply advises how to avoid contamination e.g. by centrifugation and supply information about the lid. If lids with glue or heat-seal were tested, it would be good to report the details on the lids that were used in the material section and comment on potential polymer contamination in the mass spectra.

Response: We thank the reviewer for the useful comment and we have included a detailed description and clarification in the manuscript. In brief, we used X-Pierce film to seal the plates manually as this does not take any considerable effort. The seal is only used during the digestion at 37C and for long-term peptide storage in a new sample plate. The incubation steps are performed in a PCR thermocycler with lid heating which prevents lid condensation. Upon digestion the film is carefully removed and the plate can be placed back on the Bravo for peptide recovery. We did not encounter any issues with condensation or difficulties to add or remove sealing film. We are confident to exclude this part of the workflow as a source of cross-contamination.

Having a digestion volume of <50 microliter, it is even possible to carefully vortex the plate and e.g. sonicate in a waterbath by using foam cushions that carry the plate on the water surface. However, both steps are not necessary. No centrifugation steps are necessary, no heat resistant films are necessary, and we did not observe any polymer contamination in our mass spectra.

Typos:

- Page 12, line six: The degree {degree sign} symbol is missing for 95{degree sign}C
- Page 25, second paragraph, third line: EMA \diamond EMEA
- Page 32: Hematoxylin and Eosin is written with a different font than the rest of the text and 'Hematoxylin' is written in lower case and 'Eosin' in upper case
- Figure 5: The annotation of the different tumor regions is written in capitals in A and lower case in B and D

Response: These text-issues have now been resolved.

2nd Editorial Decision

4th September 2019

Thank you for your message asking us to reconsider our decision regarding your manuscript MSB-19-9111. I have now read the referee reports once again and I have also considered your point-by-point response.

As I had outlined in my initial editorial decision letter, the most substantial concern was the lack of a fully automated workflow. Based on your point by point response, I think that the clarifications provided and the additional analyses performed (i.e. automated processing of three different tissue types) seem promising for addressing this concern. Moreover, I appreciate the clarifications regarding the novel aspects of the workflow, the description of its advantages compared to existing approaches and the additional information regarding the robustness of the analyses. Overall, I think that the proposed revisions sound reasonable and since it seems that you have most of the new data at hand, I would invite you to submit a revised version of your study (**within three months**).

Reviewer #1:

In this manuscript from Mueller and colleagues (hereafter referred to as the Mueller Report), the authors advance their solid-phase protein extraction and digestion method to an automated liquid-handling robot and validate its performance. This "automated SP3" sample preparation is then challenged with low sample inputs (protein from ~100 cells) and a real-world application in profiling protein abundance differences from human patient-derived FFPE samples of lung adenocarcinoma tumors of various histological subtypes.

This is a timely, informative and useful study that is sure to help researchers in the field of quantitative proteomics work more efficiently, reproducibly and easily on their research projects. It's pretty well written, although I have a few minor suggestions to streamline the text a bit. Although I hate to bring up the dreaded "conceptual innovation" topic, it should be clear that while incredibly useful and artfully described and validated, this is not a novel or new idea implemented for the first time. In my view, however, the utility of these methods far outweighs the often overwrought emphasis on novelty.

Response: We thank the reviewer for his/her positive assessment, seeing the potential of automated SP3 to enhance reproducible data generation in proteomics.

To enhance innovation, we have extended autoSP3 to also include tissue lysis in an integrated workflow. Therefore, **we have included new data (new Figure 5)** in the revised manuscript showing the processing of fresh-frozen tissue samples all the way to peptides, eliminating all manual handling steps, i.e. now including an automated tissue lysis step that seamlessly interfaces with automated SP3. Specifically, we collected three types of tissue (liver, kidney, heart) that were lysed in an automated and highly controlled manner using a Covaris ultrasonicator, enabling processing of 96 samples simultaneously. These samples were next transferred to the Bravo robot for automated reduction/alkylation and SP3 all the way to peptides, directly followed by LCMS analysis (**see new Figure 5**). **This demonstrates that tissue-proteomics can be performed in an automated manner without manual handling steps.**

There is considerable effort taken in the early part of the manuscript to describe things that were attempted to be automated but failed. This reads a little too conversational; if space considerations are important, much of this could be streamlined. A more succinct depiction of the workflow and the automation in the early Results section might be better served in the main text, with anecdotes reserved for maybe Supporting Information, if at all.

Response: We understand the reviewer's point, and we have made changes as suggested to achieve a more succinct description of the workflow. However, we have kept some of the description as we think it is important to understand the details of the protocol. In addition, implementation of some handling steps is not completely trivial, requiring special care when transferring the manual protocol to a robotic system. We believe that pointing this out will be useful for potential trouble-shooting or for execution of SP3 on other liquid handling platforms.

To readers not familiar with the initial SP3 publication, comparison of an automated with manual SP3 method may not seem rigorous enough. Some additional discussion (or a simple comparison experiment) on traditional FASP/8M urea/etc. preparation and digestion approaches might be helpful here (even self-citations to the 2014 paper). For example, one might expect 1ug of HeLa cell digest analyzed on a Q-Exactive HF to yield more than ~1,600 protein quantifications across a 35-minute gradient. Being not as familiar with LFQ in MaxQuant as other quantification methods, it's unclear from the text how many unique peptide and protein identifications this corresponds to. In our experience,

an instrument/sample combination such as the authors' should identify roughly 4K - 5K proteins under these conditions, which raises questions about losses during SP3 (automated or otherwise).

Response: As this is not a primary method development or method comparison article, we have not detailed all proof-of-concept experiments regarding the performance of SP3, which we investigated in more detail in our earlier work (Hughes et al., MSB 2014, Hughes et al., Nat Protocols 2019). In addition, SP3 has been compared to other widely accepted methods (e.g. FASP, iST) by several independent groups (e.g. Sielaff et al., JPR 2017, Dagley et al., JPR 2019). However, we have tried to clarify this in the text by citing published research that demonstrates the performance and benefits of SP3 compared to other methods.

We assume that the second part of the reviewer's comment refers to the results shown in **Figure 2C**. Here, we focused on proteins that have been identified and quantified without missing values across each individual day (n=10 per day). We like to point out that this is only a subset of the total number of identified proteins. The overall number of quantified proteins per sample is provided in **Supplementary Figure 3A** with an average of 3191 proteins. The data are in range with our quality control (QC) for the setup. Yet, since **Figure 2C** raised confusion for all 3 reviewers, we have revised it to enhance clarity, now including all data with at least three valid LFQ values across all 10 samples per da. All CVs remain <15%, hence our conclusions remain unaffected.

With regard to potential protein loss during SP3, we like to point the reviewer's attention to **Figure 3A**. Here, we injected the equivalent of 1 microgram protein from the four highest starting materials (1.25 µg, 2.5 µg, 5 µg, and 10 µg). The resulting plateau of equal numbers of quantified proteins and summed intensities indicates a comparable recovery and also shows a range of ~2000 quantified proteins from 1 µg, indicating efficient protein capture and recovery even for the highest sample loads (10 µg).

Finally, we thank the reviewer for the useful suggestion to compare performance of manual and automated SP3. We have now included new data comparing protein recovery by manual and automated SP3 for a 10 µg HeLa sample, followed by LCMS as a read-out. As shown in a **new Supplementary Figure 1**, this indicates very similar performance of either procedures (or even slightly better recovery for automated SP3).

It is interesting that LFQ seems to normalize some of the variability of the authors' workflow. In Figure 4 for example, the number of LFQ protein groups scales very nicely with input - but the sum of the LFQ intensities does not (see e.g. 156ng and 100cell samples). I guess it's possible that some proteins exhibit dynamic recoveries during SP3, or that losses for some proteins are absolute at or below a certain threshold of abundance. In any case, some discussion of this phenomenon is warranted, perhaps in the context how LFQ functions.

Response: We agree and we thank the reviewer for pointing this out: indeed LFQ includes a normalization step which is not intended for quantifying proteins from sample loadings that differ by orders of magnitude. The algorithm assumes that the total protein amount within each sample is identical or at least similar. Here, this is of course not the case and **we have updated Figure 3 (previously Figure 4)** and the corresponding text according to the reviewer's suggestion. Specifically, we changed LFQ to intensity-based absolute quantification (iBAQ) values to avoid any normalization across samples, now producing the expected trend.

In any case, based on our own experience (Hughes et al., MSB 2014, Hughes et al., Nat Protocols 2019) and the studies of other independent groups (e.g. Sielaff et al., JPR 2017, Dagley et al., JPR 2019) there

is no indication of dynamic or selective recovery of particular proteins, which is one of the hallmarks of SP3.

The analysis of human ADC tumor samples is a useful example of the approach. In summary, this is a useful approach that is expected to complement existing methods and add significantly to the toolbox of sample handling procedures for the field. A few minor tweaks will hopefully serve to further elevate an already nice body of work.

We thank the reviewer for the useful comments to improve our manuscript.

Reviewer #2:

Muller et al describe the automation of their previously described SP3 protocol on the Bravo liquid handling system from Agilent. They performed a detailed analysis of the reproducibility of the sample preparation by comparing intra- and inter-day variability and investigated the sensitivity by reducing the sample amount to only few ng starting material. To show the applicability of the automated workflow to clinical sample analysis, they followed with a study of NSCLC FFPE tumor sections. They analyzed the differences between regions with distinct histopathological subtype within single tumors, and characterized the functional associations of these differences. The entire study is based on the SP3 method, developed by the same lab, and already successfully implemented by many other labs in the field. The application of the method to FFPE was also shown before by Hughes (previously in the Krijgveld lab). Overall, this is a nicely presented study, and the analyses are very solid. However, despite the importance of automation, I think it is a rather incremental technological development. Nevertheless, the additional Bravo protocols will surely be useful to the community. There are several points that still require some attention:

1. The entire manuscript is focusing on the reproducibility of the analysis, and shows the CVs of commonly identified proteins. It is critical to assess the reproducibility of the identification rate, how many missing values, the association of missing values with intensity etc. They refer to missing values only in a single sentence (p26), which is insufficient to evaluate the robustness of the analyses.

Response: We believe that there must have been a misunderstanding since commonly identified proteins were solely considered in **Figure 2C**, aimed to indicate the CVs of commonly identified and quantified proteins within each day. Here, the relative differences in the comparison of median and average CVs between days is relevant rather than the absolute levels of CV percentages. However, since **Figure 2C** raised confusion for all 3 reviewers, we have revised it to enhance clarity, now including all data with at least three valid LFQ values across 10 samples per day. In the **revised Figure 2C**, all CVs remain <15%, hence our conclusions remain unaffected.

In all other analyses, we had focused on proteins with an LFQ intensity in at least 45 out of 60 raw files (75% data completeness, see **new Supplementary Figure 3B**), covering 79.04% of the entire list of proteins in the dataset. In addition, we had reported the median and average CVs observed from the complete list of quantified proteins (n=3750) across all 60 samples to be 18.1% and 20.5%. To make this explicit and to avoid further confusion, **we have extended Table 1** to include CVs for the complete list of proteins with minimum three valid values across all samples.

We agree with the reviewer that, beyond CVs, identification rates and missing values are important parameters for proteomics experiments. These data are in fact provided in the manuscript, although admittedly a bit hidden. For instance, the identification rate per sample is provided in **Supplementary Figure 3A**. In addition, the effect of protein abundance on CV follows from **Supplementary Figure 5**.

We had not included data on the association of missing values with intensity, since this follows the expected trend that proteins of low abundance are identified with a higher number of missing values (see **Revision Figure 1** below). This is typical for any shotgun proteomics experiment, and does not inform on the reproducibility of the experiment. Therefore, we think there is no need to include **Revision Figure 1** in the manuscript since it does not provide insight in the performance of autoSP3.

Revision Figure 1. Correlation between protein intensities and missing values.

2. The ability to recover proteins from very low amounts of input material is impressive. However, they indicate the starting amount and the number of identified proteins, which largely depends on the LC-MS parameters/performance. It would be more informative to know the exact recovery from each amount of starting material. They should quantify the amount of peptides before loading on the HPLC (at least for the larger amounts that are within the detection range of common quantification methods).

Response: We agree with the reviewer that a quantification of peptide amounts prior to the injection would be informative. However, we would like to note two points: i) as the reviewer already indicated, for the majority of samples no common quantification method (peptide assay) would be sufficiently sensitive to generate convincing data. ii) it is nearly impossible to compare a sample before and after SP3 by MS. In particular, it would require the removal of SDS from the sample not treated by SP3 by an alternative method. Even if this can be achieved, the comparison would be between SP3 and the alternative method, not between a sample before and after SP3.

If the main interest or concern of the reviewer is the general recovery of peptides from a given protein input during the SP3 protocol, this has been extensively validated in our previous work (Hughes et al.,

MSB 2014, Hughes et al., Nat Protocols 2019) and by other independent groups (e.g. Sielaff et al., JPR 2017, Dagley et al., JPR 2019). To indicate to the reviewer that performance of automated SP3 is equivalent to manual SP3, we have compared recovery by manual and automated SP3 for a 10 µg HeLa sample, indicating very similar performance, now included as a **new Supplementary Figure 1**.

3. The clinical sample analyses show higher levels of ECM proteins in the lepidic samples. Similar protein families are also shown for the papillary subtype. The string network shows also higher levels of stromal components, such as immune cells, IFN-related proteins, neutrophil markers etc. These results suggest that the tissue dissection didn't sample the actual cancer cells, but more of the stroma. It is therefore not convincing that these differences represent the biology of these subtypes, and not a sampling artifact. In order to have better assessment of the biological relevance of their findings, they can try to correct for tumor sample cellularity. A better approach could analyze the histology. I understand that the manuscript is focused on the technology and not tumor biology and therefore understand if these analyses are beyond the scope of this study. Nevertheless, the authors should at least refer to these issues in writing.

Response: We agree with the reviewer's concern about tumor cell content (TCC). We therefore added a **new Supplementary Figure 6D** complementary to **Figure 4B**, now color-coded for TCC in each sample. Since TCC distribution is random, this indicates that the separation is not driven by the TCC. In the manuscript, we have now pointed out the potential influence of TCC.

4. In the analysis of the clinical samples, the authors define significantly changing proteins as having $p\text{-value} < 0.05$ (methods and figure). I assume that they couldn't find truly significant proteins (with FDR correction), but if so, they should not refer to those as significant.

Response: we agree that we should have included the p-value correction from the beginning. We have now done this, and we have updated the corresponding parts of the manuscript (Text, Methods, and Figures).

5. It is not clear how the enrichment analysis was performed and what background was used. The String database usually uses the entire genome/proteome as background, which is not the appropriate background when the coverage is rather low with clear biases against lowly expressed proteins. Instead, they should use the entire identified dataset. In addition, the presentation of the enrichment analyses in the Supplementary Figures 9 and 10 is not clear. Is each category only significantly enriched in one direction? Are there categories that are de-enriched? Maybe a different visualization would be clearer.

Response: We added a clarification regarding the STRING-based gene ontology enrichment analyses. In brief, we used the entire dataset as a specific background consisting of 5642 proteins. In addition, we required highest confidence (minimal required interaction score of 0.9) and excluded text mining-based evidence. Disconnected nodes in the network analyses were hidden. All remaining settings were left at default.

6. Supplementary Figure 3 indicates #LFQ intensities, should be corrected to #LFQ proteins.

Response: this has now been corrected.

7. Tables at the end of the submitted pdf should be removed.

Response: we will clarify this with the handling editor and adhere to MSB guidelines.

Reviewer #3:

Reviewer comments on the manuscript `Automated sample preparation with SP3 for low-input clinical proteomics`

General remarks

Jeroen Krijgsveld's laboratory invented the SP3 protocol several years ago (Hughes et al., MSB 2014) and the protocol itself is state-of-the-art and easy to use, which resulted in broad adaption in the proteomics community. Moreover, the mechanism of this protocol, which is based on protein aggregation on sub-micron particles, has been described recently (Batth et al, MCP, 2019), making the protocol even more trustworthy. The manuscript describes the automation of parts of this protocol and the authors did an extensive evaluation of its analytical performance with intra-day variation, inter-day variation analyses and cross-contamination tests. The availability of the software and the fact that the Agilent Bravo systems are widely spread in laboratories can result in a broad adaption of this robotic workflow.

The application to minimal sample amounts will be interesting for readers in the proteomics community in which several labs are in a race towards single cell analysis. The tumor tissue as a low input material is a valid example, but tissue slices of 5 mm x 5 mm x 5 µm are still far away from single cell analysis and can also be handled by other protocols. Nevertheless, the combination of low sample amounts and automation gives the manuscript an interesting technical aspect, which will be the largest advance of this manuscript.

Having said that the work in the manuscript was carried out well, it still lacks novelty and suffers from the fact that the workflow is only partially automated. Even though automation of sample preparation is fantastic to guarantee reproducibility in cases where large-numbers of samples are necessary to process, automation of this part of the proteomics workflow is not novel itself and has been even realized over the complete sample preparation workflow and not only starting from lysed samples. This has been facilitated before by several scientific laboratories and several companies like PreOmics and Agilent - both having a product pipeline running on the automation of the proteomic sample preparation (more information below).

The performed clinical study within the manuscript has a medium to large size for a proteomics study analyzing tissue samples and is interesting to read. The study and its analysis are performed well. The manuscript itself reads very nicely and the figures are well drafted. Overall, the authors present an interesting protocol for the partly automation of the previously described SP3 protocol, which **lacks novelty** and the **automation is overstated**. To make the protocol truly interesting for the community, the **complete workflow from the raw sample to the ready-to-inject peptides would have to be automated** without any manual interference.

Response: We thank the reviewer for his/her overall positive assessment of the manuscript, and for seeing the value of automated SP3 for the proteomics community. Yet, we partly disagree with the major criticisms with regard to lack of novelty as well as overstatement of automation. As explained in the detailed comments below, this is most likely due to an underestimation of what is included in

automated SP3, and in overestimating what other methods do. In addition, in response to the valid point that tissue lysis was not included in the automated pipeline, **we have now included new data (Figure 5)** describing the extension of autoSP3 with a **seamlessly integrated method for tissue lysis and extraction**. Collectively, this constitutes a workflow that processes tissues to peptides in a 96-plex manner, reducing manual intervention to transferring a sample plate from one device to the next (**Figure 5A**). We believe this significantly expands the innovative aspect beyond the introduction of autoSP3 alone.

Major points

1. Automation of the sample preparation step in MS-based proteomics is crucial to guarantee reproducibility for the processing of high numbers of samples. However, automation of this part of the proteomics workflow is not novel itself. This has been described and implemented in previous studies, even on the Agilent Bravo system (page 6, second paragraph). Moreover, Agilent has application notes for sample preparation workflows available and supplies the software protocol files ready-to-use (<https://www.agilent.com/cs/library/applications/5991-3602EN.pdf>; <https://www.agilent.com/cs/library/applications/application-protein-sample-preparation-preomics-bravo-proteomics-5994-0306en-agilent.pdf>). Even the complete automation of the sample preparation for various tissues is already possible. For example PreOmics sells an instrument for the automated sample preparation starting from cells/tissues to ready-to-inject samples (<https://www.preon.preomics.com/>). Furthermore, the potential to automate the SP3 protocol has been suggested before (Batth et al., MCP, 2019). Consequently, the work in this manuscript is not novel in terms of supplying an automated platform for clinical samples.

Response: We agree that several parts of the proteomic workflow have been implemented on different robotic platforms before, using various methods. The iST method, used both in the Agilent and Preomics platforms cited above, is one of them, however these procedures are neither generic nor fully automated as suggested by the reviewer, thereby contrasting to automated SP3 for various reasons:

- i) Solid-phase extraction and reversed phase chromatography, like iST, are incompatible with the use of SDS or other detergents (e.g. as is preferred for FFPE tissue, see below), thereby limiting the use of both the Agilent and Preomics protocols. In fact, the cited protocol (<https://www.agilent.com/cs/library/applications/application-protein-sample-preparation-preomics-bravo-proteomics-5994-0306en-agilent.pdf>) is a method to process plasma samples where no protein extraction is needed. We mentioned this in the main text: “Automated sample preparation is far less common in proteomics, and is restricted to cases where detergents can be avoided (iST, plasma proteomics21)”. We believe that the ability to use SDS is a very strong asset of SP3 especially for FFPE tissues: heating in SDS is the preferred way for FFPE protein extraction (e.g. Wisniewski, Dus and Mann, Prot Clin Appl 2013), pre-empting the use of iST including the protocols cited by the reviewer. This is further illustrated by the fact that the original iST paper (Kulak et al, 2014) has been cited 674 times, however none of these studies used it for FFPE tissue (based on Google Scholar: an ‘in-article’ text search for the term FFPE retrieves 15 out of 674 papers. Detailed reading these 15 papers indicated that none used iST for FFPE samples). Yet, this is not to say that iST or any other sample prep method does not have any value – quite the contrary, as we now point out more explicitly in the text. The point we want to make is that autoSP3 has non-overlapping characteristics with other methods, extending the options for experimentalists to choose their preferred methodology.

- ii) **Neither of the cited protocols include a tissue extraction step:** they both start from (detergent-free) proteins extracts from cells or tissue. Therefore, contrary to the reviewer's claim, the **Preomics protocol does NOT start from intact tissues.**
- iii) The cited Agilent protocol includes several centrifugation steps which are performed OFF the deck of the Bravo system. This contrasts with autoSP3 which is completely performed ON the deck (i.e. no manual intervention).
- iv) The Preomics instrument can process 12 samples simultaneously, compared to 96 for autoSP3.
- v) The existence of the Preomics instrument does not remove the novelty of our automated SP3 protocol, since it adds the key benefits of SP3 compared to iST as discussed in several independent studies (e.g. Sielaff et al., JPR 2017, Dagley et al., JPR 2019). In addition, the Bravo liquid handling platform is widely available in many genomics and biochemistry laboratories which facilitates adoption of the method by the proteomic community.

It is correct that the suggestion has been made before that the SP3 protocol is amenable for automation. Yet, this was first suggested by ourselves when introducing SP3 (Hughes et al., MSB 2014) and not by Batth et al (2019), since this potential was immediately clear from the inception of SP3. Furthermore, we hope that the reviewer appreciates that there is a big difference between suggesting a method, and effectively realizing and implementing it. A method to process 96 samples in parallel for low-input proteomics did not exist before, hence we believe this is an important innovation.

2. Key sample preparation steps requiring intensive hands-on time such as the processing until the lysate is generated from a cell line or a tissue sample are not automated in the manuscript. This includes sample solubilization, boiling and sonication to enable efficient protein extraction. These steps are crucial and easily subjected to proteomic variation and should be integrated in the automated pipeline.

The extensive time period for pre-heating is a valid reason for having an external device, but this only leaves the extraction of proteins from the lysate and the pipetting of enzymes as automated steps. In this manuscript more steps are manual than automated and questions the 'automated sample preparation' in the title of the manuscript. Moreover, it seems that the authors describe further hands-on steps needed for recovering and acidification of the peptides: "Following enzymatic digestion, peptide samples can be manually recovered and acidified in new plates or tubes (Figure 1G), or this can optionally be done on-deck after supplying new pipette tips". In summary, only the aliquoting of beads and the organic solvents buffer exchange during the washing steps (which is specific to this protocol) are automated. All other critical steps, including sample lysis, digestion and harvesting of the peptides are still done manually. This would strongly limit the adaption of this protocol by other laboratories.

The authors claim at several passages of the manuscript that the protocol eliminates hands-on time, which should be rectified. Eliminating of hands-on time would mean that raw samples are placed in the robot and ready-to-inject peptides are harvested afterwards without any interference from the user. This is also the impression that the authors give the reader.

Response: These concerns must be based on a misunderstanding, possibly caused in part by the diverse of workflows that are described in the manuscript where steps can be performed on or off the liquid handling platform as preferred by the user. In addition, the reviewer may underestimate the inherent simplicity of the SP3 workflow, meaning that there is a limited number of steps that need to be automated. For clarity: protocols A, B, C, and D (now **Figure 1C**, previously **Supplementary Figure 1**) include all steps for automated reduction/alkylation and SP3 in an unattended process starting from a

lysed sample to peptides ready for MS (i.e. combining reduction and alkylation, protein clean-up, digestion, and acidification). The only interference is the sealing of the sample plate and transferring it to an incubator for digestion, and moving it back to the robot for acidification (<30 seconds hands-on time). Optionally (and this may have created confusion), the user can choose to perform reduction/alkylation and acidification manually to result in a slightly faster (but interrupted) procedure. We have now clarified in the text which automated protocols are available, and which ones we used in our experiments.

The reviewer is correct that tissue lysis was not part of the automated pipeline. Therefore, we have extended autoSP3 to **also include tissue lysis in an integrated workflow**. Therefore, **we have included new data (new Figure 5)** in the revised manuscript showing the processing of fresh-frozen tissue samples all the way to peptides, eliminating all manual handling steps, i.e. now including an automated tissue lysis step that seamlessly interfaces with automated SP3. Specifically, we collected three types of tissue (liver, kidney, heart) that were lysed in an automated and highly controlled manner using a Covaris ultrasonicator, enabling processing of 96 samples simultaneously. These samples were next transferred to the Bravo robot for automated reduction/alkylation and SP3 all the way to peptides, directly followed by LCMS analysis (see **new Figure 5**). **This demonstrates that tissue-proteomics can be performed in an automated manner without manual handling steps.**

In conclusion, we believe that the combination of tissue ultrasonication and SP3, for 96 samples and without manual sample handling, provides a powerful and innovative workflow for tissue proteomics. Moreover, the ability to process tissues to peptides sets this approach apart from what can be done by the Proteomics instrument, or by previously established methods on the Agilent Bravo.

3. The experiments for assessing the intra-day and inter-day variation are valid for this propose in general but still have some experimental and analytical issues. The assessment of the CV's should be discussed in the manuscript. Clinical assays and platforms that might be suited for this process in future have to be assessed with all transparency. The protocol in the manuscript starts with samples that were already in a very advanced state of the sample preparation pipeline, which prevents the calculation of the repeatability and reproducibility of the workflow. All these steps will contribute to further variation. The authors aim to have an assessment according to clinical guidelines, but such an assessment have to include all parts of the sample preparation workflow and not only a sub process. This fact should be openly stated and discussed in the manuscript.

Response: We thank the reviewer for the remarks regarding the assessment of CVs. We clarified in the text that we are talking about the repeatability and reproducibility of the automated SP3 workflow.

The comparison of samples processed using manual versus autoSP3 clearly shows that the overall reproducibility is very high (**Figure 2D**). Considering the high throughput achieved using the automated workflow, doing this manually for 96 samples would easily take several hours of manual pipetting, which is prone to errors the longer it takes.

To clarify sample handling prior to SP3: in the case the cell dilution series (10-10,000 cells, **Figure 3B, previously Figure 4B**), cells were serially diluted into SDS-containing lysis buffer in a micro-titer plate, **and the plate (i.e. all samples simultaneously) were sonicated in a waterbath sonicator** before transfer to the Bravo robot. We realize that this was not properly described in the methods section, so we have corrected this in the revised manuscript and we articulated this in the results section. We show with 8 replicates (processed on two different plates) that our workflow is highly reproducible from beginning to end of the sample processing. The error bars indicate high consistency of the results despite processing of quantity-limited samples. Showing proteomic data for samples as small as 100 cells or less is rare in itself, doing this in an automated manner has not been done before.

In addition, for the new data that combines ultrasonication and autoSP3 (**new Figure 5**) we have determined CVs for the workflow that spans all sample handling steps, i.e. including tissue lysis and protein extraction, reduction and alkylation, protein clean-up, digestion and LCMS. This shows (**new Figure 5E**) that CVs for all tissues remain under 14%, demonstrating excellent performance. The value for HeLa cells is only slightly higher (15.5%), probably reflecting differences in manual pipetting to collect cells for the replicate experiments.

4. Focusing on proteins without any missing values is cherry-picking for the calculation of CVs. These proteins will be the highest abundant and most reproducible quantified proteins. This is also reflected by only looking at the 1650-1700 proteins from the probably 5000(?) quantified proteins in this analysis. The mass spectrometer was operated in DDA mode, selecting peptides in a semi-stochastic fashion for sequencing and missing values are parts of our research. Nevertheless, most proteins will be present at several time points and CVs can be calculated and they should not be excluded from the analysis. The transparent way for reporting CVs would mean to include all proteins with at least three values.

Response: We believe that there must have been a misunderstanding since we considered commonly identified proteins **solely in Figure 2C**. Here, the relative differences in the comparison of median and average CVs between days is relevant rather than the absolute levels of CV percentages. However, to avoid further confusion, we followed the reviewer's suggestion and included all proteins with at least three valid intensities across n=10 samples within a processing day (**revised Figure 2C**). This only marginally affected CVs (all <0.15), hence our conclusions remain valid.

In all other analyses, we had focused on proteins with an LFQ intensity in at least 45 out of 60 raw files (75% data completeness, see **new Supplementary Figure 3B**), covering 79.04% of the entire list of proteins in the dataset. In addition, we had reported the median and average CVs observed from the complete list of quantified proteins (n=3750) across all 60 samples to be 18.1% and 20.5%. To make this explicit and to avoid further confusion, **we have extended Table 1** to include CVs for the complete list of proteins with minimum three valid values across all samples.

5. In samples with limited amounts of starting material the match-between-run algorithm tends to assign peptide `identifications` to background ions. Did the authors apply the match-between-run algorithm for the low sample amounts? If this was the case, the numbers without matching should be reported and correlations to measurements with higher sample amounts should be made to supply convincing data for correct peptide assignments.

Response: In both experiments of quantity-limited input material (**now Figure 3A and 3B, previously Figure 4A and 4B**) the match between runs algorithm was **not** used for obvious reasons as indicated by the reviewer. We clarified this in the manuscript (Methods section) to avoid confusion.

6. The SP3 protocol is a fantastic proteomics sample preparation protocol. Nevertheless, throughout the manuscript other protocols are described with negative attributes that are not true. Statements like "Among the most popular sample preparation methods, stage tips, and its derivative iST, do not tolerate detergents thereby restricting their generic use." or "The ability to handle detergent-containing samples, including SDS, is a great benefit over other methods such as iST, ..." (page 36) do other state-of-the-art proteomics sample preparation technology no justice. Samples from all kinds of tissues and plant materials were processed with protocols like the iST or with the help of other solid phase-extraction protocols, resulting in the most extensive proteomes ever reported (Coon et al., Nat. Biotech., 2016; Bekker-Jensen et al., Cell Syst., 2017; Doll et al., Nat. Comm., 2017).

Strong detergents like SDS are not necessary in general and in cases where someone wants to use them, they can be removed by protein precipitation.

Response: It was not our intention to discredit any of the other methods. All of the mentioned approaches have their pros and cons and have their specific reasons to be established in the proteomic community. However, our statement points to the key properties where we believe SP3 stands out, e.g. not requiring solid-phase-extraction protocols or protein precipitation. Of course, most of the previously mentioned protocols can be combined with other methods and integrated in a compatible way to generate extensive proteomes. We have revised the text to remove the impression of discrediting previous work, while explaining assets of SP3.

It is correct that SDS is not necessary for all sample types, although it is advantageous for many. The prime reason why researchers resort to detergent-free extraction methods is that elimination of detergents is cumbersome yet necessary (e.g. for MS). SDS may be removed by precipitation, however we hope that the reviewer agrees that this is not a realistic option for low-input samples e.g. <1000 cells (**now Figure 3B**). Having SP3, researchers are free to choose whether they employ detergents, chaotropes, or other conditions for sample preparation.

Minor points

- The cited guideline for the evaluation of an analytical platform is thought to be for 'veterinary drug residue studies'. In general, it might be more appropriate to cite the FDA CLIA guidelines or recommendations from the proteomics community that are based on them (Grant and Hoofnagle, Clin Chem, 2014).

We thank the reviewer for the useful comment and we have included the reference in the manuscript.

- The timelines for the analysis as they are depicted in supplemental Fig. 1 are very interesting as they give the reader an instant understanding of the protocols and could be shown already in the main Fig. 1.

We thank the reviewer for the useful comment, and we now display it as **Figure 1C**.

- Fig. 2D: Can the authors comment on the relative poor correlation of the manual SP3 and the robotic protocol? What could be the reason for this effect? This might be important to report as this/these factor/s might have an effect on the sample preparation itself and could help to further optimize the protocol.

Response: We think that there was a misunderstanding by the reviewer, possibly caused by the colour scheme in the Figure spanning a very narrow scale bar in the very high SD-range (1-0.94), in fact indicating very high correlation between the manual and robotic protocol. As we already stated in the main text, this indicates that SP3 is highly robust in both formats (SD >0.97), where SD of 0.94 between both versions may be caused by subtle differences e.g. in sample volumes. In addition, both data sets were collected months apart, likely reflected in LCMS performance differences (e.g. different/new trap and/or analytical column, MS performance).

We have considered to change the scaling of **Figure 2D** from 1-0.94 (**Review Figure 2**: original, left panel below), to 1-0.50 (**Review Figure 2**: right panel below). Although both panels represent the exact same data, we believe that the left panel is more informative than the blue square that results from

the widened SD-window, and therefore we suggest to keep the original figure. We have adapted the text to avoid mis-interpretation of the Figure.

Review Figure 2. Pearson correlation heatmap with original (left panel) and widened SD-window from 1 to 0.50 (right panel). Please note the different scale bars.

- Page 28, paragraph 2: Please state the size of the bins e. g. 1000 proteins, 2 orders of magnitude for each bin

Response: As already highlighted within the corresponding **Supplementary Figure 5A and 5B** (previous **Figure 3A**), we clarified the size of each abundance bin in the text.

- The authors claim that the sample preparation only takes 1.5h (page 25, page 37), allowing the capacity of hundreds of samples. This is an overstatement and should be avoided as it leads to a misinterpretation for the reader (see above major point 2). The 1.5h includes only the purification of the proteins. Other sample preparation steps are neglected and they prevent that hundreds of samples can be prepared. The samples have to be lysed, RNA/DNA removed, disulfide bridges have to be reduced and alkylated (taking 45-65 minutes according to the authors) and most importantly the proteins have to be digested (2-16h; page 23). To avoid a misunderstanding the authors should additionally name the time for the complete sample preparation, when describing the whole sample preparation time and correct the potential of how many samples can be prepared.

Response: We agree that we should have been more careful making our point, and that the mentioned throughput refers to the actual SP3 processing while lysis and digestion time come on top. We have now changed this text in the light of the new data shown in **Figure 5**, where tissue lysis was included in the protocol, and where we now have actual data on how long this takes in practice. Furthermore, we have slightly toned down to what extent this can be scaled up, while still pointing out that significant throughput can be achieved with minimal intervention by the operator.

- The reviewer does not agree that "filtration, centrifugation, precipitation and electrophoresis are difficult to standardize and unsuitable for automation". Especially the first two steps are already implemented in almost all robotic platforms like the Tecan, Hamilton, or Agilent systems.

Response: We agree with the reviewer and we have made changes in the manuscript to clarify our statement. While filtration and centrifugation steps can be automated they require appropriate and diverse instrumentation.

- In the method section it would be useful to know which magnetic rack was used in the Bravo system.

Response: We thank the reviewer for pointing to this omission, we have now added the information to the method section.

- Some references e.g. 26, 38, 39 start with first names. Please review all references.

This has now been corrected.

- If a lid was necessary, how did the authors ensure that no droplets from condensate cross-contaminated the samples. Did the authors apply additional centrifugation steps e.g. to ensure that no condensate was on the lid or that air bubbles were removed from the buffers? If so, it would be helpful to inform the readers about this steps and supply advises how to avoid contamination e.g. by centrifugation and supply information about the lid. If lids with glue or heat-seal were tested, it would be good to report the details on the lids that were used in the material section and comment on potential polymer contamination in the mass spectra.

Response: We thank the reviewer for the useful comment and we have included a detailed description and clarification in the manuscript. In brief, we used X-Pierce film to seal the plates only used during the digestion at 37C and for long-term peptide storage in a new sample plate. The incubation steps are performed in a PCR thermocycler with lid-heating which prevents lid-condensation. Upon digestion the film is removed and the plate can be directly placed back on the Bravo for peptide recovery. We did not encounter any issues with condensation or difficulties to add or remove sealing film. We did not observe cross-contamination originating from this or any other part of the workflow.

Having a digestion volume of <50 microliter, it is even possible to carefully vortex the plate and e.g. sonicate in a waterbath by using foam cushions that carry the plate on the water surface. However, both steps are not necessary. No centrifugation steps are necessary, no heat resistant films are necessary, and we did not observe any polymer contamination in our mass spectra.

Typos:

- Page 12, line six: The degree {degree sign} symbol is missing for 95{degree sign}C

- Page 25, second paragraph, third line: EMA ◊ EMEA

- Page 32: Hematoxylin and Eosin is written with a different font than the rest of the text and 'Hematoxylin` is written in lower case and 'Eosin´ in upper case

- Figure 5: The annotation of the different tumor regions is written in capitals in A and lower case in B and D

Response: These text-issues have now been resolved.

Thank you again for submitting your revised work. We have now heard back from two of the three referees who were asked to evaluate your study. The reviewers are the same as those of the initial submission. Since the recommendations of both reviewers are similar, we decided to proceed with making a decision based on these two reports. If we receive comments from reviewer #2 within the next few days, I will forward them to you. As you will see below, both reviewers are supportive and recommend publication, pending a minor text modification suggested by reviewer #3.

REFEREE REPORTS

Reviewer #1:

All of my concerns have been addressed. The authors have done a fantastic job of responding to concerns of the reviewers.

Reviewer #3:

Reviewer comments on the manuscript 'Automated sample preparation with SP3 for low-input clinical proteomics'

The main criticism of the manuscript was regarding to a lack of novelty due to other automated sample preparation platforms and that the automation of the workflow in the manuscript was overstated. In the revised manuscript the authors added a high-throughput tissue lysis system that can off-line prepare 96 samples. This has not been implemented so far, adds novelty and will be very useful in future applications. Additionally, the authors added a more precise description of the necessary sample handling and analysis steps. Summarizing, the authors addressed all points sufficiently.

The numbers below refer to the numbers from the point-by-point answers.

1. Lack of novelty

Even though that I still see a lack of novelty in the automation of the SP3 workflow itself, I agree that adding tissue lysis in a 96-well format on the Covaris system makes the whole manuscript much more relevant and applicable. Tissue lysis is a crucial sample preparation step and the ability to perform it in high-throughput is definitely very valuable even though that this is not integrated in the robotic system and has to be done offline. The two references lowering the novelty (Preomics and the automated plasma proteomics protocol) do not include tissue lysis. Therefore, including now high-throughput tissue lysis together with the new Figure 5 definitely strengthen the manuscript. Exemplifying the preparation across three different tissue samples will also show the reader the broad applicability and is very convincing for me. Unfortunately, the supplied link to the PRIDE repository of this new dataset did not work in my hands.

2. Overstatement of elimination of hands-on time

Also in the revised version, I think that "elimination" of hands-on time is overstated and a better phrasing would be reducing or limiting. As the authors wrote "sealing of the sample plate and transferring to an incubator for digestion, and moving it back to the robot" are indeed some steps requiring manual interference. I completely agree that the protocol is very nice with very little hands on time and that it decreases manual interference, but it is still not eliminating the hands on time. Therefore, I would still recommend that the authors should report this clearly. I even think that pointing out the little hands-on time of 30 seconds will further increase the trustworthiness in the workflow itself.

3. CV calculation across the workflow

The additional experiments including all sample processing steps is appreciated and is evaluating the workflow sufficiently.

4. Proteins considered for calculating CVs

The revised calculation of the CVs for three valid intensities were added to the revised manuscript. Moreover, reporting the data completeness together with the CV calculation is an convincing and unbiased analysis that the reader can follow.

5. Match-between-runs

Good to point out in the revised manuscript that no match-between-runs has been applied.

6. Comparison of SP3 to other state-of-the-art protocols

The revised text is appropriate now.

Minor comments

All minor comments have been addressed by the authors.

2nd Revision - authors' response

4th December 2019

Response to Reviewer Comments:

1. 1) Regarding the elimination of hands-on time we followed the

reviewer's suggestion and toned down the corresponding parts of the manuscript. However, we want to point out that we were referring to the elimination of sample handling steps. By this, we are referring to manual handling/pipetting steps actively involving the samples, which are fully eliminated. The remaining manual interference within our workflow is restricted to moving or sealing of plates and preparing the different platforms for running. We clarified this further in the text by minor changes.

2. 2) The link to the proteomics data at PRIDE is functional.

With this we hope that we addressed all remaining editorial concerns and issues. In case that there is anything left to be done we are happy to do so. We are looking forward to sharing our work through the excellent platform of MSB. As soon as the work is published we will take care to make the proteomic data and the Bravo protocol files accessible at PRIDE.

Accepted

5th December 2019

Thank you again for sending us your revised manuscript. We are now satisfied with the modifications made and I am pleased to inform you that your paper has been accepted for publication.

YOU MUST COMPLETE ALL CELLS WITH A PINK BACKGROUND ↓
PLEASE NOTE THAT THIS CHECKLIST WILL BE PUBLISHED ALONGSIDE YOUR PAPER

Corresponding Author Name: Jeroen Krijgsveld
Journal Submitted to: MSB - Molecular Systems Biology
Manuscript Number: MSB-19-9111RR